# MA-RLHF: Reinforcement Learning from Human Feedback with Macro Actions

**Yekun Chai**[*] **Haoran Sun**[*] **Huang Fang  Shuohuan Wang  Yu Sun  Hua Wu**
Baidu Inc.
{chaiyekun,fanghuang,wangshuohuan}@baidu.com
sunhaoran0402@gmail.com

## Abstract

Reinforcement learning from human feedback (RLHF) has demonstrated effectiveness in aligning large language models (LLMs) with human preferences. However, token-level RLHF suffers from the credit assignment problem over long sequences, where delayed rewards make it challenging for the model to discern which actions contributed to preferred outcomes. This hinders learning efficiency and slows convergence. In this paper, we propose MA-RLHF, a simple *yet* effective RLHF framework that incorporates *macro actions* — sequences of tokens or higher-level language constructs — into the learning process. By operating at higher level of abstraction, our approach reduces the temporal distance between actions and rewards, facilitating faster and more accurate credit assignment. This results in more stable policy gradient estimates and enhances learning efficiency within each episode, all without increasing computational complexity during training or inference. We validate our approach through extensive experiments across various model sizes and tasks, including text summarization, dialogue generation, question answering, and program synthesis. Our method achieves substantial performance improvements over standard RLHF, with performance gains of up to 30% in text summarization and code generation, 18% in dialogue, and 8% in question answering tasks. Notably, our approach reaches parity with vanilla RLHF $1.7 \sim 2$ times faster in terms of training time and continues to outperform it with further training. We make our code and data publicly available at https://github.com/ernie-research/MA-RLHF.

## 1 Introduction

Recent advancements in large language models (LLMs) have revolutionized natural language processing tasks, demonstrating impressive capabilities across a wide range of applications such as code generation (Roziere et al., 2023; Chai et al., 2023; Lozhkov et al., 2024), mathematical reasoning (Lewkowycz et al., 2022; Anil et al., 2023), and dialogue assistance (OpenAI, 2023; Team et al., 2023; Anthropic). Despite these successes, aligning LLMs with human values and preferences remains a critical challenge. Reinforcement learning from human feedback (RLHF) has emerged as a promising approach to address this alignment issue by incorporating human evaluations into the training process (Christiano et al., 2017; Ziegler et al., 2019; Stiennon et al., 2020).

Existing RLHF (Ouyang et al., 2022; Bai et al., 2022; Askell et al., 2021) methods mainly optimize decisions at the level of individual tokens, and require to process a vast number of minute adjustments. However, this fine-grained training paradigm can lead to the credit assignment problem (Kaelbling et al., 1996; Pang et al., 2019; Machado et al., 2023b; Pignatelli et al., 2023), particularly when dealing with long-distance dependencies. As LLM agents attempt to optimize decisions across extensive sequences, the difficulty in attributing the credits of actions to specific tokens complicates the reinforcement learning (RL) process (Pignatelli et al., 2024). Moreover, the use of subword tokenization, such as Byte-Pair Encoding (Sennrich et al., 2016), often splits words into

---

[*]Equal contribution. Correspondence to: YC.
    Work done during HS's internship at Baidu.

smaller pieces. For instance, OpenAI's ChatGPT[1] treats each token as three quarters of a word on average, resulting in sequences that are 33% longer than word counts (OpenAI, 2024) and further exacerbates the credit assignment problem.

Additionally, standard RLHF methods may overlook essential local co-occurrence patterns or inherent structures between adjacent tokens in natural language. For example, consider the phrase `Big Apple`[2], treating `Big` and `Apple` as isolated decisions misses the cohesive meaning of the term, which actually refers to the "New York City". The token-level granularity of natural language can hinder the agent's ability to capture high-level language constructs in RL optimization, as some sequences are better understood when evaluated holistically.

To address these challenges, we propose a new framework called macro-action RLHF (MA-RLHF) that incorporate **macro action** — sequences of tokens or high-level language constructs — into the RLHF framework. The concept of macro actions, has been explored in the literature of planning (Iba, 1989; Korf, 1985; Sacerdoti, 1974) and reinforcement learning (Thrun & Schwartz, 1994; Precup et al., 1997; Hauskrecht et al., 2013), simplifies decision-making by operating at high levels of temporal abstraction under the framework of semi-Markov Decision Processes (SMDPs) (Sutton et al., 1999b). Macro actions leverage temporal abstraction by chunking the sequences and reducing the decision resolution, enabling the agent to learn from "long-sighted" macro-level actions instead of "short-sighted" token-level actions. This can potentially lead to improved learning efficiency and scalability. Alternatively, MA-RLHF can also be interpreted from the perspective of *reversing tokenization*; MA-RLHF serves as a de-tokenization process to reconstruct high-level language units from subword pieces. By merging tokens into macro actions, we reduce the number of decision points and shorten decision trajectories, alleviating the credit assignment problem caused by long temporal distances.

To conclude, our main contributions are as follows:

- We propose MA-RLHF, a simple *yet* effective RLHF framework that integrates the macro actions into RLHF to align LLMs with human preference. We demonstrate the effectiveness of our approach through extensive experiments across various datasets and tasks, including text summarization, dialogue generation, question answering, and code generation.
- We show that MA-RLHF achieves $1.7\times$ to $2\times$ faster learning efficiency in reward scores during training compared to the standard token-level RLHF, without introducing additional computational costs during training or inference. MA-RLHF also exhibits strong scalability across model sizes ranging from 2B to 27B parameters.
- Our analysis reveals that MA-RLHF exhibits robust generalization capabilities under varying experimental settings, such as temperature values and rejection sampling, consistently outperforms the standard RLHF approaches.

## 2 PRELIMINARIES

We introduce some basic concepts and notations used in RL and RLHF.

### 2.1 REINFORCEMENT LEARNING AND POLICY OPTIMIZATION

**Problem Definition** RL addresses the problem of finding a policy to make optimal sequential decisions in environments modeled as a Markov Decision Process (MDP) (Sutton & Barto, 1999). An MDP is defined by the tuple $(\mathcal{S}, \mathcal{A}, P, r, \rho_0, \gamma)$, where $\mathcal{S}$ denotes a finite set of states, $\mathcal{A}$ is a finite set of actions, $P : \mathcal{S} \times \mathcal{A} \times \mathcal{S} \to [0, 1]$ represents the state transition probability distribution, $r : \mathcal{S} \times \mathcal{A} \to \mathbb{R}$ is the reward function, $\rho_0 : \mathcal{S} \to [0, 1]$ defines the initial state distribution, and $\gamma \in (0, 1)$ is the discount factor that determines the importance of future rewards.

Given a trajectory $(s_0, a_0, s_1, a_1, \cdots)$, a reward $r_t = r(s_t, a_t)$ is received at each time $t$. The state-action value function $Q_\pi(s_t, a_t) = \mathbb{E}_{s_{t+1}, a_{t+1},...} \left[ \sum_{l=0}^{\infty} \gamma^l r_{t+l} \right]$ measures the expected return of taking action $a_t$ at state $s_t$ and following policy $\pi$ thereafter. The value function $V_\pi(s_t) = \mathbb{E}_{a_t, s_{t+1},...} \left[ \sum_{l=0}^{\infty} \gamma^l r_{t+l} \right]$ estimates the expected return from state $s_t$ under the policy $\pi$. The advantage function $A_\pi(s_t, a_t) = Q_\pi(s_t, a_t) - V_\pi(s_t)$ reflects the relative value of taking action $a_t$ at state $s_t$ compared to the average value of the state.

---

[1]https://platform.openai.com/tokenizer
[2]https://en.wikipedia.org/wiki/Big_Apple

The goal of RL is to find an optimal policy $\pi_\theta(a \mid s)$, parameterized by $\theta$, that maximizes the expected cumulative discounted reward: $J(\theta) = \mathbb{E}_{s_0,a_0,\dots}[\sum_{t=0}^{\infty} \gamma^t r_t]$, where $s_0 \sim \rho_0(s_0)$ represents the initial state distribution, $a_t \sim \pi_\theta(a_t \mid s_t)$ denotes the action selection based on the policy, and $s_{t+1} \sim P(s_{t+1} \mid s_t, a_t)$ specifies the state transition dynamics.

**Proximal Policy Optimization** Policy gradient methods are a common approach for optimizing policies by estimating the gradient of a performance objective with respect to the policy parameters $\theta$. The policy gradient is given by: $\nabla_\theta J(\theta) = \mathbb{E}[\sum_{t=0}^{\infty} A_t \nabla_\theta \log \pi_\theta(a_t \mid s_t)]$, where the expectation $\mathbb{E}[\cdot]$ is taken over the randomness of the initial state, policy, and state-transition. The policy gradient guides us how to adjust the policy parameters to improve the expected return. Among the family of policy gradient methods, Proximal Policy Optimization (Schulman et al., 2017, PPO) is perhaps the most widely-used one due to its simplicity and empirical effectiveness. PPO simplifies TRPO (Schulman et al., 2015) by using a clipped surrogate objective function to penalize large deviations from the old policy, thereby ensuring more stable updates. Specifically, PPO introduces a clipped objective function:

$$J^{\text{ppo-clip}}(\theta) = \mathbb{E}_t \left[ \min \left( \frac{\pi_\theta(a_t \mid s_t)}{\pi_{\theta_{\text{old}}}(a_t \mid s_t)} A_t, \text{clip}(\frac{\pi_\theta(a_t \mid s_t)}{\pi_{\theta_{\text{old}}}(a_t \mid s_t)}, 1 - \epsilon, 1 + \epsilon) A_t \right) \right], \quad (1)$$

where $\epsilon$ is a hyperparameter that defines the range for clipping. The expectation $\mathbb{E}_t[\dots]$ indicates the empirical average over a finite batch of samples. Nowadays, PPO usually comes as the first choice for RL practitioners.

## 2.2 RLHF FOR HUMAN ALIGNMENT

The post-training of LLMs (Stiennon et al., 2020; Ouyang et al., 2022) is a multi-stage training paradigm to align LLMs with human preferences. Post-training typically involves three stages:

**(1) Supervised Fine-Tuning (SFT)** stage: A pre-trained language model (LM) is fine-tuned on a dataset of human demonstrations, learning to generate responses that align with human instructions and preferences.

**(2) Reward Modeling (RM)** stage: A reward model is trained on a labeled preference dataset $\mathcal{D} = (x_i, y_i^+, y_i^-)_{i=1}^N$, consisting of prompts $x_i$ and pairs of responses $(y_i^+, y_i^-)$, where $y_i^+$ is preferred over $y_i^-$ by human annotators. The reward model $r_\phi(x, y)$, parameterized by $\phi$, is trained using the ranking loss: $\mathcal{L}_{\text{RM}} = -\log \sigma(\log(r_\phi(x, y_+) - r_\phi(x, y_-)))$, where $\sigma$ denotes the sigmoid function.

**(3) RLHF** stage: The RL fine-tuning utilizes the RM to provide feedback on the generated outputs, optimizing the policy using RL methods such as PPO. The reward signal is modified by incorporating a Kullback-Leibler (KL) divergence penalty to balance the exploration of new policies with adherence to the SFT model. The reshaped reward is defined as:

$$R(x, y) = r_\phi(x, y) - \beta D_{\text{KL}}(\pi_\theta(\cdot \mid x) \parallel \pi_{\text{sft}}(\cdot \mid x)),$$

where $\pi_\theta$ represents the policy learned through RL, $\pi_{\text{sft}}$ is the policy produced from the SFT stage, and $\beta > 0$ is a hyperparameter that controls the strength of the KL penalty.

In the RLHF stage, the PPO algorithm, as detailed in Equation (1), is employed to optimize the RL policy. In the context of RLHF, we denote the state $s_t = \{s_0, a_0, a_1, \dots, a_{t-1}\}$ as the sequence of tokens generated up to time step $t$, while $s_0$ represents the initial states, *i.e.*, the prompt, and $a_t$ represents the token selected at the $t$-th position.

# 3 MARCO-ACTION RLHF

## 3.1 REVISITING MACRO ACTIONS (OPTIONS)

**Macro actions**, also referred to as **options** (Sutton et al., 1999b), are high-level constructs that encapsulate a sequence of primitive actions (*i.e.*, subword tokens); by its definition, macro actions allows an agent to operate at a coarser temporal scale.

Formally, a macro action is characterized by three components: (1) a policy $\pi : \mathcal{S} \times \mathcal{A} \rightarrow [0, 1]$ which guides the action selection among actions; (2) a termination condition $\zeta : \mathcal{S}^+ \rightarrow [0, 1]$, which determines where the macro action should end; (3) a initiation set $\mathcal{I} \subseteq \mathcal{S}$, which is a subset of states

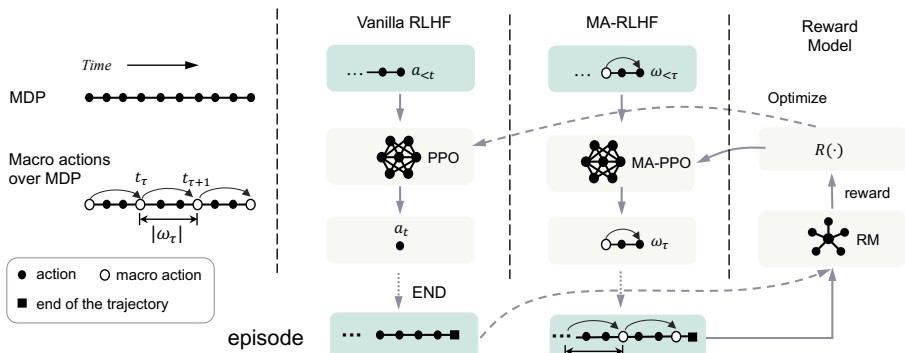

Figure 1: Illustration of the MA-RLHF optimization framework. Standard RLHF makes decisions and evaluates value scores at the token level, while MA-RLHF makes decisions over sequences of tokens at a coarser temporal scale.

that macro actions can begin with. Once initiated with a state $s_0 \in \mathcal{I}$, the macro action follows policy $\pi$ until it reaches the termination condition according to $\zeta$. Intuitively, the use of carefully designed macro actions can extend decision-making temporally, it allows the agent to avoid "short-sighted" token-level decisions and encourage "long-sighted" macro-level decisions, thereby simplifies the decision-making process and potentially enhances learning efficiency.

## 3.2 RLHF WITH MACRO ACTIONS

We describe how we integrate macro-actions into the existing RLHF framework, the resulting framework is named as macro-action RLHF (MA-RLHF).

### 3.2.1 FORMALIZATION OF MACRO ACTIONS

We denote macro actions as $\omega_1, \omega_2, \ldots, \omega_\tau$. In the context of LLMs, a macro action $\omega_\tau$ consists of a sequence of consecutive tokens, $i.e., \omega_\tau = \{a_{t_\tau}, a_{t_\tau+1}, \ldots, a_{t_{\tau+1}-1}\}$, where $t_\tau$ is the starting index of the $\tau$-th macro action. We let $|\omega_\tau|$ denotes the number of primitive actions that $\omega_\tau$ contains. Unless otherwise specified, we use $\tau$ to index macro actions/states and use $t$ to index primitive actions/states.

As mentioned in §3.1, macro actions are defined by the policy model, the termination condition and the initiation set. In MA-RLHF, we set the policy model the same as the standard token-level RLHF and let the initiation set to be any possible sequence of tokens. Therefore, the macro action used in MA-RLHF is decided solely by the termination condition, which plays a crucial rule in the MA-RLHF framework. We explore three termination conditions in this work:

- $n$-**gram based termination**: Following Vezhnevets et al. (2016), we find that $n$-grams serve as a simple *yet* effective termination condition for macro actions, $i.e., |\omega_\tau| = n$, where $n$ represents the length of the $n$-gram. We consider two variants of the $n$-gram termination condition: (a) **Fixed $n$-gram**: We group tokens into fixed-length $n$-grams, simplifying the action space while maintaining common linguistic patterns. We empirically find fixed $n$-gram macro action perform best and use it as the default setup. (b) **Randomized $n$-gram**: We randomly select the length of a $n$-gram from a predefined list of lengths $n \in \{2, 3, 5, 10\}$ to introduce variability, allowing the policy to adapt to different sequence lengths.

- **Parsing-based termination**: $\omega_\tau$ is derived from syntactic or semantic parsing of the input text, aligning macro actions with grammatical structures like phrases or clauses. Concretely, we traverse the constituent tree of the entire sequence using depth-first search (DFS), expanding non-terminal nodes until current non-terminal state contains no more than a specified threshold of leaf tokens, set at $C = 5$.

- **Perplexity-based (PPL) termination**: Perplexity measures the likelihood of a sequence of tokens. Here, the perplexity of a macro action is proportional to the averaged entropy of the token within it, $i.e., \text{ppl}(\omega_\tau) \propto -\frac{1}{|\omega_\tau|} \sum_{a \in \omega_\tau} \log p_a$. A macro action terminates until it reaches a token that has negative impact on the perplexity of the macro action. Mathematically, we construct $\omega_\tau = \{a_{t_\tau}, \ldots, a_{t_{\tau+1}-1}\}$ such that $\text{ppl}(\omega_\tau \cup a_{t_{\tau+1}}) > \text{ppl}(\omega_\tau)$ and $\text{ppl}(\{a_{t_\tau}, \ldots, a_i\}) \geq \text{ppl}(\{a_{t_\tau}, \ldots, a_{i+1}\})$ for all $t_\tau \leq i \leq t_{\tau+1} - 2$.

After determining the macro action based on the termination condition, we apply the state value function and importance sampling at the macro level Equation (1). We provide the details of implementation in Appendix D.1.

### 3.2.2 POLICY OPTIMIZATION WITH MACRO ACTIONS

In MA-RLHF, we adapt the PPO algorithm for optimization, referred to as MA-PPO. In the context of LLMs, expanding the action space with additional macro actions/tokens results in re-architecting the LLM's vocabulary and retraining the model, which is computationally prohibitive. Thus, we maintain the original action space as pretrained LLMs, which can be treated as "single-step" primitive options as noted in (Sutton et al., 1999b). The policy $\pi_\theta$ still outputs probabilities over individual tokens, but for optimization, we consider the joint probability of the macro action: $\pi_\theta(\omega_\tau \mid s_\tau) = \prod_{t=t_\tau}^{t_{\tau+1}} \pi_\theta(a_t \mid a_{<t})$. The macro reward for executing the macro action $\omega_\tau$ at the macro time step $\tau$ is defined as: $R_\tau = \mathbb{E}\big[ \sum_{i=0}^{|\omega_\tau|-1} \rho^i r_{t_\tau+i} \mid s_\tau \big]$, where $r_t$ is the reward received at time step $t$, and we set the discount factor $\rho = 1$ in our experiments.

Each macro action represents a contiguous sequence of tokens, and is treated as an option in the SMDP framework. The option-level value function with macro action is then estimated as:

$$V^\pi(s_\tau, \omega_\tau) = \mathbb{E}\left[ R_\tau + \gamma V^\pi(s_{t_{\tau+1}}) \mid s_\tau, \omega_\tau \right],$$

where $\gamma$ is the discount factor for future rewards beyond the macro action.

The advantage function $A_\pi(s_\tau, \omega_\tau)$ in MA-PPO determines how much the chosen macro action outperforms the average, which is defined as $A_\pi(s_\tau, \omega_\tau) = Q_\pi(s_\tau, \omega_\tau) - V^\pi(s_\tau)$. Similar to the definition stated in §2, $Q_\pi(s_\tau, \omega_\tau)$ is the expected return conditioned on executing $\omega_\tau$ at state $s_\tau$, which is calculated by summing the immediate macro rewards from the macro action with the discounted value of the subsequent state.

In MA-PPO, the objective function is adapted for MA-level evaluation. The policy gradient is computed based on the advantage of the MA sequences:

$$\mathcal{L}^{\text{MA-PPO}}(\theta) = \mathbb{E}_\tau \left[ \min \left( \frac{\pi_\theta(\omega_\tau \mid s_\tau)}{\pi_{\theta_{\text{old}}}(\omega_\tau \mid s_\tau)} \hat{A}_\tau, \; \text{clip} \left( \frac{\pi_\theta(\omega_\tau \mid s_\tau)}{\pi_{\theta_{\text{old}}}(\omega_\tau \mid s_\tau)}, 1 - \epsilon, 1 + \epsilon \right) \hat{A}_\tau \right) \right],$$

where $\hat{A}_\tau$ is the estimated advantage at macro time step $\tau$, $\epsilon$ is a constant that defines the range for clipping, and $\pi_{\theta_{\text{old}}}$ is the policy before the update.

### 3.2.3 CONNECTION TO PREVIOUS METHODS

MA-RLHF builds on and generalizes prior work in the RLHF literature by varying the length of macro actions. When the macro action length is set to 1, MA-RLHF reduces to the standard token-level RLHF (Stiennon et al., 2020; Ouyang et al., 2022), operating as an MDP. Conversely, if we allow $|\omega_\tau| \to \infty$, then MA-RLHF converges toward methods like RLOO (Ahmadian et al., 2024), REINFORCE (Williams, 1992; Sutton et al., 1999a), and GRPO (Shao et al., 2024), approximating a contextual bandit problem where decisions are made based on the entire sequence context. By varying the length of macro actions $|\omega_\tau|$, MA-RLHF provides a flexible framework that balances the granularity of action decisions. We provide further analysis on the impact of $|\omega_\tau|$ in §4.3.

## 4 EXPERIMENTS

### 4.1 EXPERIMENTAL SETTINGS

**Tasks and Datasets** We evaluate MA-RLHF on three different datasets for open-ended generation tasks: TL;DR (Stiennon et al., 2020) dataset for text summarization, Anthropic Helpful and Harmless (HH-RLHF) (Bai et al., 2022) for dialogue generation[3], and WebGPT Comparison (Nakano et al., 2021) for question answering. Additionally, we evaluate MA-RLHF on code generation using the APPS (Hendrycks et al., 2021) dataset. More details can be found in Appendix B.1.

---

[3] https://huggingface.co/datasets/Dahoas/full-hh-rlhf

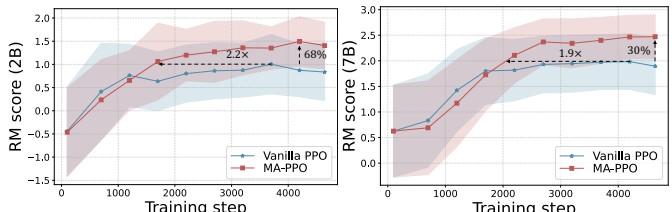 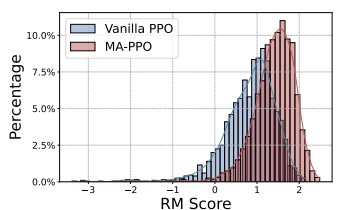

Figure 2: Test RM scores of Gemma-2B and Gemma-7B models on the TL;DR dataset. The shaded regions represent the standard deviation on test RM scores across training runs.

Figure 3: RM score distribution for PPO and MA-PPO (2B) at final steps (4.6k) on TL;DR.

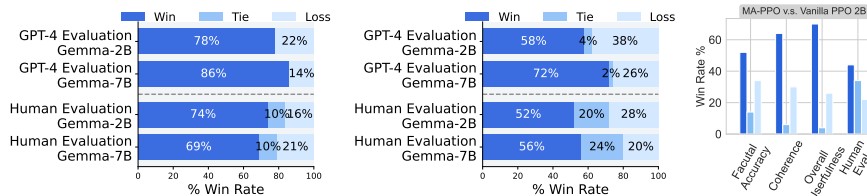

Figure 4: Win rates of MA-PPO against vanilla PPO on **TL;DR** (left), **HH-RLHF** (middle) and **WebGPT Comparisons** (right), estimated by GPT-4 and Human.

**Base Models and Training Details**  For open-ended generation tasks, we use pre-trained Gemma-2B (Team et al., 2024) as our base model; we further adopt Gemma-7B and Gemma-2-27B to test the scaling trend. For the program synthesis task, we use CodeGemma-1.1-2B and CodeGemma-1.1-7B-it as our base models. The data split for SFT / RM / PPO and the hyperparameters used in SFT / RM / PPO stages are detailed in Appendix B.2. The implementation details of MA-PPO can be found in Appendix E.

**Evaluation**  For open-ended generation tasks, our evaluation metrics includes RM scores, GPT-4 pairwise evaluation, and human pairwise evaluation. To compute the RM score, we randomly sample 2k validation instances for the TL;DR and HH-RLHF datasets and use the default validation set of the WebGPT dataset. For GPT-4 and human evaluations, we simulate the win-rate on 50 instances that are drawn from the instances used in the RM evaluation. The GPT-4 and human evaluations are based on task-specific criterion: relevance, coherence, consistency, and fluency for TL;DR; help-fulness for HH-RLHF; factual accuracy, coherence, and usefulness for WebGPT. We followed prior studies (Askell et al., 2021; Zheng et al., 2024) by randomizing the order of responses during evaluation to mitigating potential evaluation biases. The prompts used by the GPT-4 evaluation are placed in Appendix F.1, and the annotation rules used for human evaluation are given in Appendix F.2. For the program synthesis task, we utilize pass@1 and pass@5 metrics to assess the performance of the model, evaluated on the provided 5k test set.

## 4.2 MAIN RESULTS

In this section, we present the main results of applying MA-PPO across three key tasks: summarization, dialogue, and question answering. The main takeaway is that MA-PPO consistently outperforms vanilla PPO in terms of both training efficiency and generation quality; MA-PPO obtains a significant improvement in testing reward model scores and human/GPT-4 evaluation win rates.

**TL;DR Summarization**  For the TL;DR summarization task, MA-PPO shows a marked improvement over vanilla PPO. As shown in Figure 2, MA-PPO achieves parity with vanilla PPO approximately $1.7 - 2$ times faster during training. Specifically, Gemma-2B trained with 1.7k MA-PPO updates reaches similar testing RM scores obtained by vanilla PPO trained with 3.7k steps. We also find similar trends when scaling up the parameter sizes to 7B, demonstrating the generalized capability of MA-PPO on model sizes.

Moreover, Figure 3 highlights the distribution of RM scores, where MA-PPO consistently shifts towards higher RM sores compared to vanilla PPO. Further evaluation using GPT-4, given in the left figure of Figure 4, shows that MA-PPO achieves 78% and 86% win rate over vanilla PPO for the 2B and 7B models, respectively. Human evaluation gives similar results, where MA-PPO obtains win rates of 74% and 69%, further demonstrating the effectiveness of macro actions. The final testing RM scores of MA-PPO and vanilla PPO are given in Table 2.

Table 1: Agreement among RM, GPT-4, and human evaluations on TL;DR.

|  | #Param | RM | GPT-4 | Human |
|---|---|---|---|---|
| RM |  | 100% | - | - |
| GPT-4 | 2B | 78% | 100% | - |
| Human |  | 76% | 58% | 100% |
| RM |  | 100% | - | - |
| GPT-4 | 7B | 78% | 100% | - |
| Human |  | 74% | 64% | 100% |

Table 2: Test RM scores of vanilla PPO and MA-PPO on TL;DR, HH-RLHF, and WebGPT datasets.

| Model | TL;DR | HH-RLHF | WebGPT |
|---|---|---|---|
| Vanilla PPO (2B) | 0.84 | 1.31 | -0.62 |
| MA-PPO (2B) | $1.41_{+68\%}$ | $1.55_{+18\%}$ | $-0.60_{+3\%}$ |
| Vanilla PPO (7B) | 1.90 | 1.05 | -0.61 |
| MA-PPO (7B) | $2.47_{+30\%}$ | $1.24_{+18\%}$ | $-0.56_{+8\%}$ |

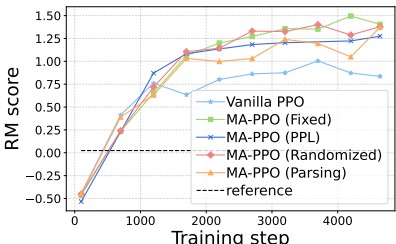
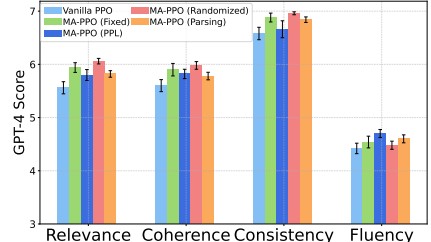

Figure 5: Performance of MA-PPO with various macro action termination strategies on the TL;DR dataset using Gemma-2B. **Left:** Test RM scores for different termination strategies. **Right:** GPT-4 evaluation across four dimensions – relevance, coherence, consistency, and fluency – comparing different MA termination methods.

**HH-RLHF Dialogue**  We use the HH-RLHF dataset to evaluate the helpfulness and harmlessness of single-turn dialogues. MA-PPO shows clear advantages over vanilla PPO, as depicted in the middle figure of Figure 4. GPT-4 evaluations show that MA-PPO yields a 72% win rate for the Gemma-7B model, compared to 58% for the Gemma-2B model. Human evaluation results align with these findings, with the win rate increasing from 52% to 56% as model size scales from 2B to 7B. The testing RM score of MA-PPO and vanilla PPO are presented in Table 2. These results highlight the scalability and effectiveness of MA-PPO in dialogue tasks. We refer to Appendix C.1 for detailed experimental results.

**WebGPT Comparisons**  We evaluate MA-PPO on the WebGPT Comparison dataset for question-answering tasks. As shown in Figure 4 (Right), MA-PPO consistently outperforms vanilla PPO, with GPT-4 evaluations yielding a win rate of 64% for the Gemma-7B model. This result demonstrate the robustness of MA-PPO across different tasks, including more structured tasks like question answering. More experimental details refer to Appendix C.2.

**Validating Model-based Judgments with Human Evaluation**  We evaluate the reliability of our evaluation methods by calculating the agreement between the reward model, GPT-4, and human evaluators. Since GPT-4 and human evaluations are conducted pairwise, we determine the reward model's win rate by selecting the summary with the higher RM score. The results, shown in Table 1, demonstrate that the reward model aligns more closely with both GPT-4 and human evaluations. Furthermore, the agreement between GPT-4 and human evaluators averaged 62% across models, reinforcing the consistency and validity of our evaluation framework.

### 4.3 ANALYZING THE USE OF MACRO ACTIONS

We study the performance of various termination strategies. Unless otherwise specified, we conduct our analysis on the TL;DR dataset.

#### 4.3.1 EXPLORING DIFFERENT STRATEGIES FOR MA TERMINATION ($\zeta$)

In MA-RLHF, the termination condition ($\zeta$) for macro actions is critical as it determines when a macro action should conclude. We compare the performance of various termination strategies, particularly on reward maximization and linguistic coherence. The termination strategies studied in this section including fixed / randomized $n$-gram-based, parsing-based, and perplexity-based termination, as aforementioned in §3.2.1; please see Figure 12 for detailed illustration.

Figure 5 illustrates the overall test-set performance on RM scores (Left) and GPT-4 evaluation scores (Right) with different MA termination strategies. All macro action termination strategies outperform the vanilla PPO approach, underscoring the importance of temporal abstraction in decision-making. Figure 5 (Left) shows that $n$-gram based approach, both fixed and randomized, achieves the opti-

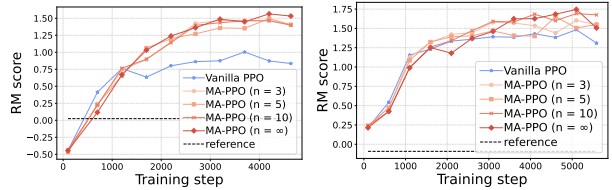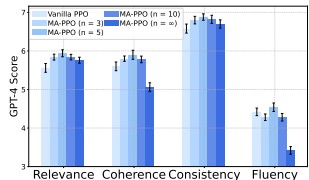

Figure 6: Test RM scores of different $n$ values in MA-PPO evaluated by corresponding RM on the TL;DR (left) and HH-RLHF (right) dataset.

Figure 7: GPT-4 scores of vanilla PPO and MA-PPO with different $n$ values on TL;DR.

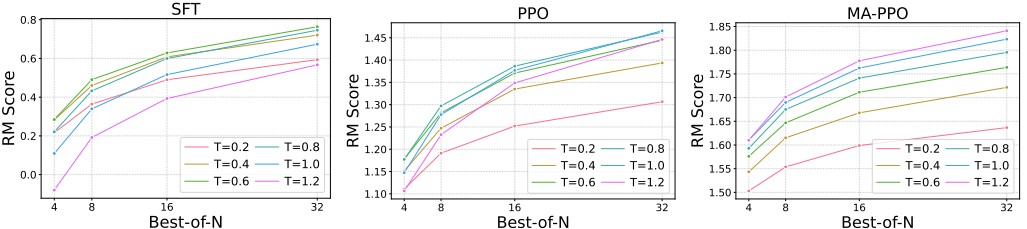

Figure 8: The effect of temperature on RM scores for varying sample sizes (Best-of-$N$) across models. (**Left**): RM score of the **SFT** model under different temperatures and sample sizes. (**Mid**): RM score of **vanilla PPO** under the same settings. (**Right**): RM score of **MA-PPO**.

mal results among others. Notably, randomized $n$-gram-based termination performs the best across multiple dimensions, including relevance, coherence, and consistency, as shown in Figure 5 (Right). As expected, the perplexity-based termination enhances fluency, and is most suited for tasks that prioritize smooth and natural language generation. Furthermore, parsing-based termination shows promising ability to handle complex grammar, as it is designed to better capture linguistic structures.

### 4.3.2 ABLATION STUDY: VARYING $n$ IN MA-RLHF

The $n$-gram based macro action strategy in MA-RLHF uses a hyper-parameter $n$ to control the length of macro actions. Notably, when $n = 1$, MA-PPO is equivalent to vanilla PPO, and treats the problem as a traditional Markov Decision Process (MDP), making decisions token by token. In contrast, setting $n \to \infty$ corresponds to the REINFORCE algorithm (McGovern & Sutton, 1998), where the entire sequence is treated as a single macro action, akin to a contextual bandit problem, as discussed in § 3.2.3. For intermediate values of $n$ (*i.e.*, $n \in (1, \infty)$), MA-PPO falls under the SMDP framework, which allows for temporally extended actions; see §3. This continuum between MDPs and contextual bandits highlights the flexibility of the MA-RLHF approach in handling varying levels of temporal abstraction.

**RM Scores** We conducted experiments with varying values of $n$ ($n \in \{3, 5, 10, \infty\}$) on the TL;DR and HH-RLHF datasets. Figure 6 shows that all values of $n$ lead to performance improvements over the vanilla PPO ($n = 1$), indicating the advantage of modeling sequences of tokens as macro actions. Notably, for the TL;DR dataset, $n = \infty$ yields the highest RM score, suggesting that treating the entire sequence as a macro action is particularly effective for the summarization task. For the HH-RLHF dataset, setting $n = 10$ gives the best performance, likely because this task benefits from moderate-length macro actions that can capture essential linguistic structures while maintaining sufficient granularity.

**GPT-4 Evaluation Analysis** As shown in Figure 7, setting $n = 5$ strikes a good balance between relevance, coherence, consistency; it outperforms both smaller and larger values of $n$. These findings align with the semi-MDP framework: increasing $n$ allows for better credit assignment and context retention, but excessive abstraction (*e.g.*, $n = \infty$) sacrifices fine-grained control. Overall, moderate values of $n = 5$ and $n = 10$ provide the best trade-offs, highlighting the adaptability across tasks.

### 4.4 GENERALIZATION PROBING IN MACRO ACTIONS

**Robustness on Rejection Sampling vs. Temperature** Best-of-$N$ (*a.k.a*, rejection sampling) (Touvron et al., 2023) enhances response quality by selecting the highest-reward response from $N$ samples generated by the policy model. We compare MA-PPO, SFT, and vanilla PPO using the best-of-$N$ sampling across various temperatures $T \in \{0.2, 0.4, 0.6, 0.8, 1.0, 1.2\}$ and sample sizes $N \in \{4, 8, 16, 32\}$. As shown in Figure 8, best-of-$N$ sampling improves RM scores for all methods,

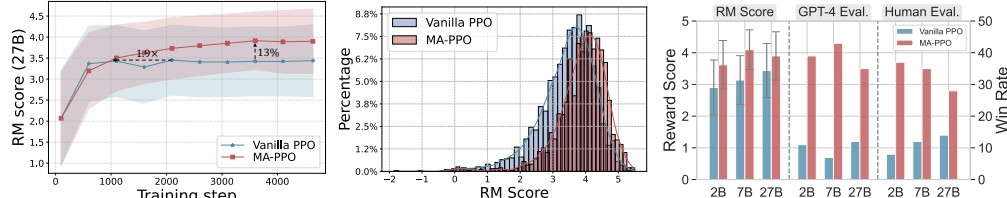

Figure 9: Evaluation results for vanilla PPO and MA-PPO on Gemma-2-27B using the TL;DR dataset. **Left**: RM scores on validation set. **Mid**: Distribution of RM scores for vanilla PPO and MA-PPO (27B) at final steps (4.6k). **Right**: Scaling trending on TL;DR dataset across 2B, 7B, and 27B model size, showing RM scores, GPT-4 evaluation, human evaluation results.

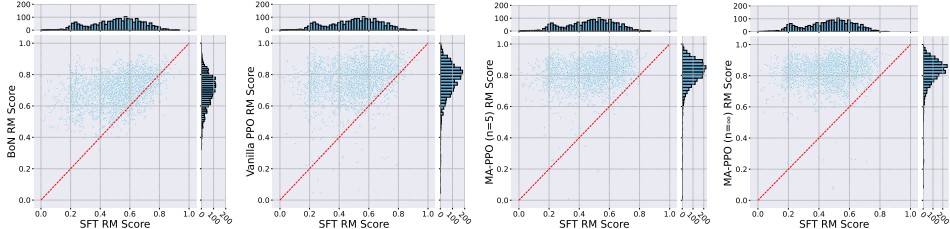

Figure 10: RM score shifting pattern after RLHF training; **Left**: RM scores of best-of-$N$ ($N = 8$) sampling compared to the SFT model. **Mid Left**: RM scores of vanilla PPO compared to the SFT model. **Mid Right**: RM scores of MA-PPO ($n = 5$) compared to the SFT model. **Right**: RM scores of MA-PPO ($n = \infty$) compared to the SFT model.

with performance increasing as $N$ grows. We observe that SFT and vanilla PPO are sensitive to temperature variations, requiring specific adjustments to achieve optimal results. In contrast, MA-PPO demonstrates robustness in sampling temperature, it consistently delivers the best performance at $T = 1.2$ and shows consistent improvement across all tested temperatures. Moreover, MA-PPO maintains stable performance across varying temperature settings, as detailed in Appendix D.4, highlighting its robustness and generalization capabilities under different sampling temperatures.

**Scaling Trends up to 27B Models** We evaluate the performance of MA-PPO across different model sizes, specifically Gemma-2B, 7B, and 27B. As demonstrated in Figure 9 (Left and Mid), MA-PPO consistently surpasses vanilla PPO, exhibiting higher RM scores throughout training. Figure 9 (Right) presents the scaling trend of MA-PPO across the 2B, 7B, and 27B models in terms of testing RM scores, GPT-4, and human evaluations. The experimental results underscore the scalability and robust performance of MA-PPO across varying model sizes.

**Analyzing the Impact on RM Score Distribution** We evaluate the RM score distribution shift after applying RLHF using vanilla PPO and MA-PPO on the TL;DR dataset, with the SFT model serving as the baseline. To further contextualize the impact of RLHF, we include the Best-of-$N$ sampling ($N = 8$) on the SFT model. As illustrated in Figure 10, Best-of-$N$ enhances overall response quality but falls short compared to RLHF. While vanilla PPO shifts the distribution towards higher RM scores, it leaves a significant number of low-quality, long-tailed instances. In contrast, MA-PPO demonstrates a more pronounced positive impact, effectively reduces the number of low-quality outliers and improves overall score distribution compared with the vanilla PPO. This highlights the robustness of MA-PPO in enhancing response quality through RLHF.

## 4.5 ADDITIONAL ANALYSIS

**Impact on $L_2$-Norm of Advantage and Q Values** We present the $L_2$-norm of both the advantage and Q-values for MA-PPO and vanilla PPO during training in Figure 11. The advantage function, which reflects the difference between the expected return (Q-value) and the baseline, is critical in guiding policy optimization. A lower $L_2$-norm of both the advantage and Q-values suggests more stable and less noisy policy updates, likely contributing to faster learning speed observed in §4.2.

The policy gradient for a sequence of length $T$ is given by: $\nabla_\theta J = \mathbb{E}\big[\sum_{t=1}^T \nabla_\theta \log \pi_\theta(a|s) \cdot R\big]$, where $R$ is the sequence reward provided by the RM. In the case of using $n$-gram based macro actions, the sequence length is reduced by a factor of $n$, shortening the decision horizon: $T \to T/n$. This reduction in the number of actions, $T/n$, where $n > 1$, implies that the temporal distance between actions and corresponding rewards is decreased, thus reducing the variance in the gradient

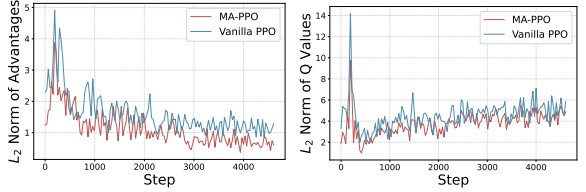

Figure 11: $L_2$ Norm of advantages and Q-values during training for MA-PPO and vanilla PPO. **Left**: $L_2$ norm of advantages over training steps; **Right**: $L_2$ norm of Q-values.

Table 3: Pass@$k$ ($k = \{1, 5\}$) metric evaluated on the APPS test set.

| Method | | CodeGemma-2B | | CodeGemma-7B | |
|---|---|---|---|---|---|
| | | PPO | MA-PPO | PPO | MA-PPO |
| pass@1 | Inter. | 2.82 | **3.25**₊15% | 4.26 | **6.22**₊46% |
| | Intro. | 15.26 | **16.56**₊8% | 20.90 | **26.74**₊28% |
| | Comp. | 0.92 | **0.94**₊2% | 1.21 | **2.00**₊65% |
| | All | 4.92 | **5.45**₊11% | 6.98 | **9.48**₊35% |
| pass@5 | Inter. | 4.10 | **4.37**₊7% | 6.57 | **8.37**₊27% |
| | Intro. | 17.30 | **18.30**₊6% | 23.30 | **30.30**₊30% |
| | Comp. | **1.70** | 1.60₋6% | 2.30 | **3.30**₊43% |
| | All | 6.26 | **6.60**₊5% | 9.06 | **11.74**₊30% |

estimate and improving credit assignment. We refer readers to Mann & Mannor (2014) for the theoretical foundations of variance reduction through macro actions and their benefits in RL.

**Case Study** We show some qualitative examples in Appendix G.1, demonstrating that MA-PPO can produce more coherent and contextually appropriate responses compared to vanilla PPO, capturing both short/long-term dependencies effectively.

**Extended Experiments: Code Generation** We further assess the effectiveness of MA-PPO on the code generation task. Following Shojaee et al. (2023); Liu et al. (2023), we utilize the compiler signal as the final reward; see Appendix B.5 for implementation details. We compare the performance of MA-PPO and vanilla PPO using the pass@k (k=1, 5) metric (Chen et al., 2021) on the 5k test set of the APPS dataset (Hendrycks et al., 2021). As shown in Table 3, MA-PPO significantly outperforms vanilla PPO in both pass @ 1 and pass @ 5 metrics, with more pronounced improvements as model size scales. Notably, for the 7B model, MA-PPO achieves an improvement of +35% in pass@1 and +30% in pass@5 over vanilla PPO, demonstrating the effectiveness of our approach in code generation tasks.

## 5 RELATED WORK

**LLM Alignment** RLHF have shown impressive success in aligning LLMs with human preferences through multi-stage training, including SFT, RM, and RL fine-tuning (Ziegler et al., 2019; Stiennon et al., 2020; Ouyang et al., 2022; Sun et al., 2025). Recent research has explored optimization methods for RL in LLMs, employing both online (Ahmadian et al., 2024; Farebrother et al., 2024; Shen et al., 2024; Chakraborty et al., 2024; Shao et al., 2024) and offline RL algorithms (Snell et al., 2023; Hu et al., 2023; Yu et al., 2024) to address training instability, improve efficiency (Tang et al., 2024) and diversity (Sun et al., 2025). Improvements to RM learning have been proposed, such as parameter scaling (Gao et al., 2023), fine-grained reward (Wu et al., 2023), tool use (Li et al., 2024), and model merging (Ramé et al., 2024; Rame et al., 2024). Alternatively, direct policy optimization (Rafailov et al., 2024; Ethayarajh et al., 2024; Gheshlaghi Azar et al., 2023; Rosset et al., 2024) has emerged as a promising approach, bypassing the instability of RL while directly aligning models to human preferences. In this paper, we enhance the RLHF action space by integrating macro actions, a well-established concept in RL (Sutton et al., 1999b; Mann & Mannor, 2014).

**Macro Action in RL** Macro actions introduce temporal abstraction in RL by grouping sequences of primitive actions, reducing decision complexity and improving long-horizon credit assignment (Precup et al., 1997; Hauskrecht et al., 2013; Sutton et al., 1999b; Pignatelli et al., 2024; Machado et al., 2023a). This method has demonstrated its utility in speeding up convergence and stabilizing policy updates in various domains (Mann & Mannor, 2014; Solway et al., 2014). Our work applies macro actions to RLHF in LLM training, leveraging this structure to enhance scalability and optimize credit assignment over extended sequences.

## 6 CONCLUSION AND FUTURE WORK

In this paper, we introduced MA-RLHF, a novel framework that incorporates macro actions into RLHF to enhance the alignment of LLMs with human preferences. Our approach demonstrates consistent improvements across multiple tasks, including summarization, dialogue generation, question answering, and code generation. Notably, MA-RLHF achieves parity with vanilla RLHF 1.7x to 2x faster in reward scores without incurring additional computational overhead, showing robust scalability across model sizes ranging from 2B to 27B parameters. It is promising to explore MA-RLHF in complex step-by-step reasoning tasks for future research.

REPRODUCIBILITY STATEMENT

We are committed to ensuring the reproducibility of the experiments presented in Section 4. To this end, we make the source code and model checkpoints publicly available at `https://github.com/ernie-research/MA-RLHF`. The detailed source code for training and evaluating both the conventional RLHF and our proposed MA-RLHF approach is included in the supplementary materials. We believe that these efforts will enable researchers to rigorously verify our findings and build upon our work.

ACKNOWLEDGMENTS

We would like to express our gratitude to the anonymous reviewers for their insightful and constructive feedback.

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

## A  LIMITATIONS

While our work demonstrates the effectiveness of MA-RLHF across multiple tasks, there are several limitations that leave room for future improvements. In our implementation, we apply the identical action / vocabulary space as pretrained LLMs, considering the fact that defining macro actions as one options (*e.g.*, one macro action per $n$-gram) would require re-architecting the LLM's vocabulary and retraining the model, which is computationally infeasible. Meanwhile, our macro action termination methods are rule-based, including linguistics- or perplexity-driven approaches; future research could explore more complex or learnable termination strategies to further enhance performance. Furthermore, regarding the generalization of MA-RLHF, our experiments are conducted using models with up to 27B parameters; exploring more advanced models, such as LLaMA 3.1 405B (Dubey et al., 2024) or other state-of-the-art architectures and tasks (*e.g.*, mathematical and complex reasoning), may provide additional insights into the scalability of MA-RLHF. Lastly, although we observe significant improvements in training efficiency, further investigation into the trade-offs between training stability and performance under diverse real-world conditions is necessary. Addressing these limitations will pave the way for more robust applications of MA-RLHF.

## B  EXPERIMENTAL DETAILS

### B.1  DATASETS AND TASKS

**TL;DR Summarization**   In this task, the policy is asked to generate summarizations for Reddit posts. This dataset consists of 93k human-annotated preference pairs and 86k pairs for validation. The trainable pairs are derived from the Reddit TL;DR (Völske et al., 2017) dataset. Additionally, a portion of the validation pairs is sourced from the CNN Daily Mails, which serves as the test set for out-of-distribution generalization.

**HH-RLHF**   With the Anthropic HH-RLHF dataset, the policy is asked to generate a helpful and harmless response given a single-turn dialogue or multi-turn dialogue. This dataset provides 112k preference-labeled instances for training, and 12.5k for validation.

**WebGPT Comparisons** The WebGPT Comparisons dataset contains QA pairs from the ELI5 (Fan et al., 2019) and the TriviaQA (Joshi et al., 2017). The policy is responsible for information retrieval and response generation. In our experimental setup, we focus exclusively on the generation task. The policy must generate a response that balances factual accuracy and coherence. This dataset contains 19.6k instances for training. We split 5% instances for validation, as no separate validation set is provided.

**Code Generation** For this task, we leverage the APPS dataset, which contains 5k training and 5k validation instances. The policy must write executable code based on a natural language described in the question, using Python as the target programming language.

We present the data statistics in Table 4.

Table 4: Statistics of datasets involved in experiments. The number of tokens are calculated with Gemma-2B tokenizer.

| Dataset | Num. of Comparisons | Num. of Train Samples | Num. of Test Samples | Avg. Tokens in Prompt | Avg. Tokens in Chosen | Avg. Tokens in Rejected |
|---------|---------------------|-----------------------|----------------------|-----------------------|-----------------------|-------------------------|
| Anthropic HH-RLHF | 127.5k | 112k | 12.5k | 160 | 83 | 75 |
| OpenAI Summarization | 179k | 92.9k | 86.1k | 325 | 35 | 33 |
| OpenAI WebGPT | 19.6k | 18.5k | 979 | 49 | 149 | 137 |
| APPS | 10k | 5k | 5k | 453 | 203 | - |

## B.2 TRAINING DETAILS

Following the procedure used by InstructGPT (Ouyang et al., 2022), we fine-tune both the SFT model and the reward model on the same dataset to avoid a distribution gap. We implement our training code with the Deepspeed-Chat package (Yao et al., 2023).

**SFT Training** We split the dataset into three parts, allocating 20% of the data in the supervised fine-tuning stage. We use the prompts and the chosen sentences as the instruction data. For the TL;DR Summarize dataset, we concatenate the post and summarization following the approach of Stiennon et al. (2020). For the single-turn dialogue and the question answering dataset, we apply a human-assistant chat template to format the instructions. For the program synthesis dataset, we format the instruction data in line with Hendrycks et al. (2021).

**Reward Modeling** In this stage, we use 40% of the data to train the reward model for each dataset, formatting the preference data the same way as in the SFT training stage. We initialize the reward model using the fine-tuned SFT model. Due to the lack of preference pairs in the program synthesis dataset, this stage is omitted for this task.

**PPO Training** Similar to previous stages, the remaining 40% of the data is used to optimize the policy model. The SFT model initializes the policy model, and the reward model initializes the critic model. For the program synthesis dataset, 80% of the data is used in this stage, with both the policy and critic models initialized using the SFT model. The pass@1 metric serves as the reward signal for program synthesis, compensating for the absence of a reward model. While training 7B model on TL;DR dataset using MA-PPO, we encountered unstable training with a KL coefficient of 0.05. Reducing the coefficient to 0.01 for the 7B model led to more stable optimization.

Table 5 lists the hyperparameters used across all training stages for each task.

## B.3 NOTATIONS

In Table 6, we present the notations used in our paper.

## B.4 DETAILS OF MACRO ACTION TERMINATION

The general form of the segmentation rule is thus $t_{\tau+1} = t_\tau + |\omega_\tau|$, where $|\omega_\tau|$ is determined by the chosen criterion, such as $n$-grams, random, parsing, or perplexity-based segmentation.

1. **Fixed $n$-gram length**: For all macro actions, we set $|\omega_\tau| = n$, where $n$ is a constant value.

Table 5: Hyper-parameters for training Gemma series of models in MA-PPO and vanilla PPO.

| Hyper-Parameter | | Gemma | | | CodeGemma | |
|---|---|---|---|---|---|---|
| | | **2B** | **7B** | **27B** | **2B** | **7B** |
| **SFT** | Batch size | 64 for WebGPT 512 for others | 128 | 128 | 16 | 32 |
| | Epochs | 3 | 5 for WebGPT 1 for others | 3 | 1 | 1 |
| | Learning rate | 1e-4 for WebGPT 5e-5 for others | 2e-5 | 5e-6 | 5e-6 | 2e-6 |
| | LR scheduler | cosine | cosine | cosine | cosine | cosine |
| | Warmup ratio | 0.1 | 0.1 | 0.1 | 0 | 0 |
| **RM** | Batch size | 32 for WebGPT 64 for others | 128 for TL;DR 64 for HH-RLHF 32 for WebGPT | 128 | - | - |
| | Epochs | 1 | 1 | 1 | - | - |
| | Learning rate | 2e-5 for WebGPT 1e-5 for others | 1e-6 | 8e-6 | - | - |
| | LR scheduler | cosine | cosine | cosine | - | - |
| | Warmup ratio | 0.1 | 0.1 | 0.1 | - | - |
| **PPO** | Batch size | 256 | 256 | 256 | 16 | 16 |
| | Policy learning rate | 1.5e-5 | 1e-6 | 7e-7 | 5e-7 | 5e-7 |
| | Critic learning rate | 1.5e-5 | 1e-6 | 1e-6 | 5e-5 | 5e-5 |
| | Epochs | 4 for WebGPT 1 for others | 4 for WebGPT 1 for others | 1 | 1 | 1 |
| | PPO epochs | 1 | 1 | 1 | 1 | 1 |
| | Rollout | 1 | 1 | 1 | 1 | 1 |
| | Clip ratio | 0.2 | 0.2 | 0.2 | 0.2 | 0.2 |
| | $\lambda$ in GAE | 0.95 | 0.95 | 0.95 | 0.95 | 0.95 |
| | $\gamma$ in GAE | 1 | 1 | 1 | 1 | 1 |
| | KL coefficient | 0.05 | 0.1 for WebGPT 0.05 for others | 0.1 | 0.05 | 0.05 |
| | Max prompt length | 512 | 512 | 512 | 600 | 600 |
| | Max response length | 512 | 512 | 512 | 512 | 512 |
| | Warmup steps | 200 | 200 | 0 | 20 | 20 |
| | Temperature | 0.8 | 0.8 | 0.8 | 1.0 | 1.0 |
| | Top-$p$ | 1.0 | 1.0 | 1.0 | 1.0 | 1.0 |
| | Top-$k$ | 50 | 50 | 50 | 5 | 5 |

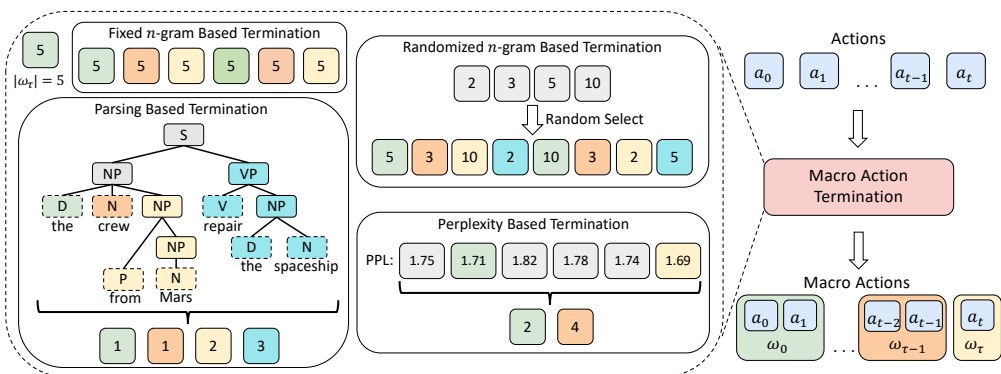

Figure 12: Illustration of four termination rules for macro actions in the MA-RLHF framework. Each termination rule outputs a list of $|\omega_\tau|$. In the parsing based termination, the macro action is determined when the token number of the current node is less than $C = 4$, which is represented as a number in the tree node.

2. **Randomized $n$-gram length**: We define a list of $\{|\omega_\tau|\} = \{2, 3, 5, 10\}$ to model macro actions. This list is repeated multiple times to cover the length of the sample, in practice, we repeat this list 3 times. If the total length of macro actions can not match the number of tokens, a large number will be considered as an additional $|\omega_\tau|$ to mitigate this gap, which is similar to the $|\omega_\tau| = \infty$. We shuffle the list and take this as a random-based length.

3. **Parsing-based length**: We parse the response into a constituent tree and perform a depth-first search (DFS) to identify macro action length. Two rules guide the termination of $|\omega_\tau|$: (1) nodes

Table 6: List of notation used in this paper.

| Sym. | Meaning |
|---|---|
| **RL** | |
| $\mathcal{S}$ | A finite set of states. |
| $\mathcal{A}$ | A finite set of actions. |
| $P$ | The state transition probability distribution. |
| $r$ | The reward function. |
| $\rho_0$ | The initial state distribution. |
| $\gamma$ | The discount factor related with future rewards. |
| $\pi_\theta(a \mid s)$ | Policy parameterized by $\theta$. |
| $\eta(\pi)$ | The expected cumulative discount reward. |
| $a_t$ | The actions selected by the policy. |
| $Q_\pi(s_t, a_t)$ | The state-action value function. |
| $V_\pi(s_t)$ | The state value function. |
| $A_\pi(s_t, a_t)$ | The advantage function. |
| $G_t$ | The expected return. |
| **RLHF** | |
| $r_\phi(x, y)$ | The reward model parameterized by $\phi$. |
| $x$ | Prompt. |
| $y^+$ | Chosen response. |
| $y^-$ | Rejected response. |
| $\beta$ | KL coefficient. |
| $\eta$ | The range for clipping in PPO. |
| $t$ | Time step of tokens. |
| **Macro Action** | |
| $\zeta$ | Termination condition. |
| $\mathcal{I}$ | Initiation set. |
| $\tau$ | The index of macro action/state/reward. |
| $\omega_\tau$ | Macro action at time step $\tau$. |
| $t_\tau$ | Time step of macro actions. |
| $\sigma_\tau$ | The weight used to measure the value of macro action. |

with fewer than $C$ tokens mark the end of a macro action; (2) nodes with single token are included in the last macro action, avoiding single-token termination conditions like punctuation. Due to differences between the training and parsing tokenizers, we revert to the standard PPO method when discrepancies occur. We set the cut-off threshold $C = 5$, providing optimal granularity in practice.

4. **Perplexity-based length**: Given a response $y$ generated by policy model, we calculate the perplexity $p_t$ at any time step $t$ by treating $y_{\leq t}$ as the ground truth response. This process leverages the logits from the reference model, avoiding additional forward passes. Intuitively, selecting the macro actions based on perplexity $\mathcal{P} = \{p_0, p_1, \ldots, p_{|y|}\}$ can be defined as selecting tokens which consistently attribute to the decrease of the perplexity given partial sentence. Mathematically, it can be represented as $\omega_\tau = \{a_{t_\tau}, a_{t_\tau+1}, \ldots, a_{t_\tau+|\omega_\tau|-1}\}$ where $\mathcal{P}_{t_\tau} = \{p_{t_\tau}, p_{t_\tau+1}, \ldots, p_{t_\tau+|\omega_\tau|-1}\}$ exhibits a monotonic decreasing pattern.

## B.5 TRAINING SETTINGS OF PROGRAM SYNTHESIS

Defining the reward score solely based on the state "Accept" or "Wrong Answer" is somewhat restrictive, as some generated code may pass certain unit tests while failing others. These actions should also receive positive signals to encourage the policy to maximize the number of passed unit tests. To address this, we incorporate an adaptive compiler signal into the reward feedback as previ-

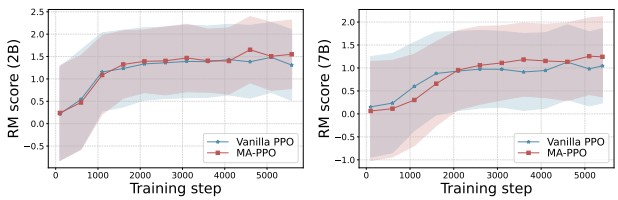 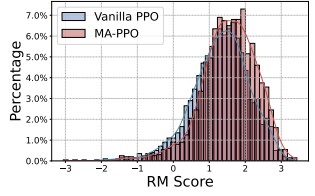

Figure 13: Test RM scores evaluated by corresponding reward model of Gemma-2B and Gemma-7B model on HH-RLHF dataset.

Figure 14: Distribution of test RM scores for vanilla PPO and MA-PPO (2B) at final steps (5.6k) on the HH-RLHF dataset.

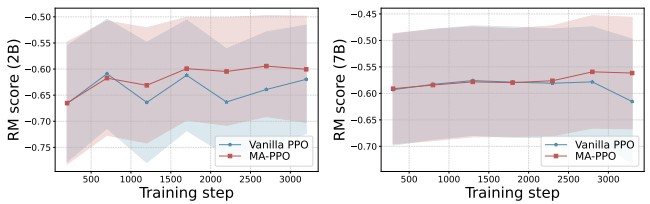 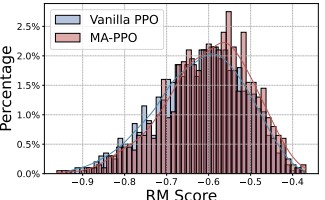

Figure 15: Test RM scores evaluated by corresponding reward model of Gemma-2B and Gemma-7B model on the WebGPT Comparisons dataset.

Figure 16: Distribution of test RM scores for vanilla PPO and MA-PPO (2B) at final steps (3.2k) on WebGPT dataset.

ous work (Shojaee et al., 2023; Liu et al., 2023):

$$R(x,y) = \begin{cases} -0.3 + 1.3 \cdot \dfrac{N_{\text{pass}}}{N_{\text{pass}} + N_{\text{fail}}}, & \text{if } y \text{ successfully compiled.} \\ -0.6, & \text{if } y \text{ received runtime error.} \\ -1.0, & \text{if } y \text{ received compile rrror.} \end{cases}$$

where $x$ represents the prompt, and $y$ represents the code snippet generated by the policy model.

## C  ADDITIONAL EXPERIMENTS RESULTS

### C.1  RESULTS OF DIALOGUE GENERATION

In Figure 13, we demonstrate the RM scores on the validation set of vanilla PPO and MA-PPO. It shows that MA-PPO surpasses vanilla PPO under RM evaluation, MA-PPO achieves parity performance at 3100 step and 2600 step for 2B and 7B models, respectively, while vanilla PPO at 5100 step and 5400 step. Generally, MA-PPO is 1.6-2x faster than vanilla PPO. Figure 14 compares the RM score distribution of both methods.

### C.2  RESULTS OF QUESTION ANSWERING

We assess the performance of MA-PPO on the OpenAI WebGPT Comparison dataset, which focuses on the question answering task.

Figure 15 presents the evaluation results based on the reward model. We observe that the policy model is challenging to optimize in this task, likely due to the suboptimal performance of the reward model. We applied early stopping during PPO training since the policy model exhibited reward hacking behavior which generated repetition tokens to inflate higher reward scores towards the end of training. Despite this, evaluations on the saved checkpoints show that MA-PPO still outperforms vanilla PPO across both tested model sizes. The reward score distribution in Figure 16 further confirms that MA-PPO achieves superior reward scores.

Table 7: Test RM scores of SFT model, vanilla PPO, MA-PPO, and baselines: DPO and RLOO on TL;DR and HH-RLHF datasets.

| Method | RM Score (TL;DR) | RM Score (HH-RLHF) |
|---|---|---|
| SFT | -0.64 | 0.13 |
| DPO | 0.03 | 0.64 |
| RLOO | 0.81 | - |
| PPO | 0.83 | 1.31 |
| MA-PPO (n=5) | **1.40** | **1.55** |

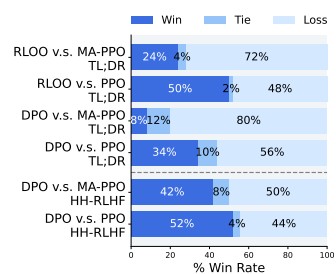

Figure 17: Win rates of DPO and RLOO against PPO and MA-PPO on TL;DR and HH-RLHF estimated by GPT-4.

When using GPT-4 as the judge, we consider three different metrics to evaluate the answers generated by the policy: factual accuracy, coherence, and usefulness overall, following previous work (Nakano et al., 2021). The win rates depicted in Figure 4 (Right) show that MA-PPO consistently outperforms the policy trained with vanilla PPO across all criteria. Notably, MA-PPO achieves higher win rates in coherence and usefulness compared to factual accuracy. Human evaluation was conducted to select the preferred answer between those generated by the two policy models. Results in Figure 4 (Right) show that answers produced by MA-PPO were predominantly preferred by human annotators.

## C.3 COMPARING WITH ADDITIONAL BASELINES

In this section, we compare MA-PPO with two additional baselines: DPO (Rafailov et al., 2024) and RLOO (Ahmadian et al., 2024) on Gemma-2B model. Both of the methods are implemented with Deepspeed-Chat. Specifically, DPO models are trained on TL;DR and HH-RLHF datasets, with the same data split as we used when training PPO. RLOO model is trained on TL;DR dataset only, with the same policy and reward model initialization as PPO. For the training details of DPO, the learning rate is set to 2e-7, with $\beta = 0.1$ for TL;DR and $\beta = 0.01$ for HH-RLHF. The policy and reference models are initialized using the same SFT model as in PPO. For RLOO, the learning rate for the policy model is set to 1.5e-5, and the number of online samples is $K = 4$. All other hyperparameters are kept consistent with PPO.

We demonstrate the results evaluated by reward model score in Table 7, and win rates estimated by GPT-4 in Figure 17. On TL;DR dataset, DPO fails to gain improvement compared to PPO and MA-PPO, while RLOO achieves similar performance compared to PPO, but outperformed by MA-PPO. On HH-RLHF dataset, DPO exhibits superior performance than PPO but still underperforms the MA-PPO.

## C.4 EXPERIMENTS ON LLAMA-3.2-3B

We conduct experiments on Llama-3.2-3B model to validate the generalizability of our method across different model families. The experiments are conducted on TL;DR dataset, following the same data split as Gemma-2B. We set the learning rates of actor and critic to 5e-6 and 1e-5, and the KL coefficient is set to 0.1. Table 8 demonstrate the results evaluated by RM score, we show MA-PPO still remark-

Table 8: Test RM scores of Llama-3.2-3B models on TL;DR dataset.

| Method | RM Score (TL;DR) |
|---|---|
| SFT | 2.38 |
| PPO | 3.33 |
| MA-PPO (n=5) | **3.96** |

ably outperforms vanilla PPO. Using GPT-4 to assess the win rate, MA-PPO obtains 61% win, 4% tie and 34% loss rate compared against PPO. These results prove the generalizability of our method.

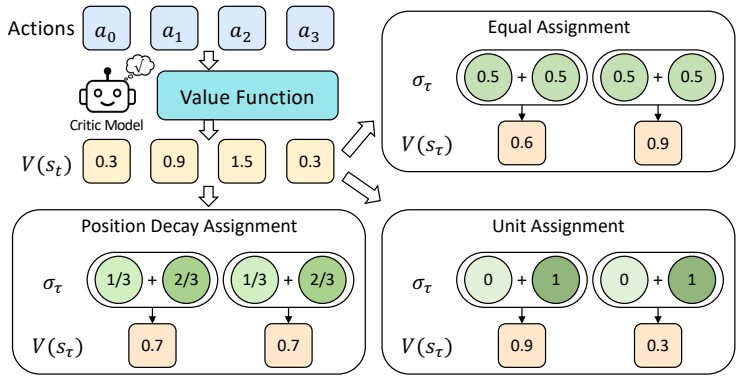

Figure 18: Illustration of value function of macro actions in MA-RLHF framework. It takes the outputs from the value function of tokens as input, and returns the value of macro actions with different $\sigma_\tau$ assignment.

Table 9: Pass@1 metric evaluated when applying different termination conditions on APPS dataset.

| Dataset | Termination | RM Score | GPT-4 Win Rate (v.s. PPO) |
|---|---|---|---|
| TL;DR | Fixed 5-gram | **1.40** | **78%** |
| | Parsing | 1.37 | **78%** |
| | PPL | 1.27 | 72% |
| HH-RLHF | Fixed 5-gram | 1.55 | 58% |
| | Parsing | **1.64** | **62%** |

Table 10: Test RM scores and GPT-4 win rates when applying different termination conditions on TL;DR and HH-RLHF datasets.

| Termination | | Fixed 10-gram | Parsing | PPL |
|---|---|---|---|---|
| pass@1 | Inter. | **3.25** | 3.17 | 3.04 |
| | Intro. | 16.56 | **17.05** | 16.36 |
| | Comp. | 0.94 | **1.24** | 0.80 |
| | All | 5.45 | **5.56** | 5.26 |

# D FURTHER ANALYSIS

## D.1 VALUE FUNCTION ESTIMATION OF MACRO ACTION

When implementing the macro actions, the value function of macro actions is estimated through the value function of tokens. This process can be formulated as: $V^\pi(s_\tau, \omega_\tau) = \sum_{i=0}^{|\omega_\tau|} \sigma_{t_\tau+i} V^\pi(s_{t_\tau+i}, a_{t_\tau+i})$, where $\sigma_\tau = \{\sigma_{t_\tau}, \cdots, \sigma_{t_\tau+|\omega_\tau|}\}$ control the contribution of each value function of tokens.

In this section, we explore several assignments of $\sigma_\tau$ and their effectiveness on MA-PPO. Figure 18 illustrates macro action value function with different $\sigma_\tau$ assignments:

1. **Equal assignment**: We treats the contributions of each value function of tokens equally when considering the value function of macro actions, *i.e.*, $\sigma_\tau = \{\frac{1}{|\omega_\tau|}\}_{i=1}^\tau$. This is the naive assignment in MA-PPO used in all our experiments.

2. **Unit assignment** Since a macro action is a higher-level construct of a sequence of actions, we can use the value function of the last action as the macro action's value function, where $\sigma_\tau = \{0, 0, \cdots, 0, 1\}$.

3. **Position decayed assignment** The contributions of each value function of tokens are determined by taking the position into consideration. We define $\sigma_\tau$ based on the position of the token, *i.e.*, $\sigma_\tau = \{\frac{1}{(|\omega_\tau|-i)\cdot\mathcal{H}}\}_{i=0}^{|\omega_\tau|-1}$, where $\mathcal{H} = \sum_{i=0}^{|\omega_\tau|-1} \frac{1}{(|\omega_\tau|-i)}$, this construction ensures $\sum_{\sigma \in \sigma_\tau} \sigma = 1$.

We tested these approaches with fixed $n$-gram based termination on TL;DR dataset, with $n = 5$. We report the RM score and GPT-4 score as previous. Results in Figure 19 show that the equal assignment yields higher RM scores. However, the unit assignment achieves the best consistency and fluency according to GPT-4 evaluations.

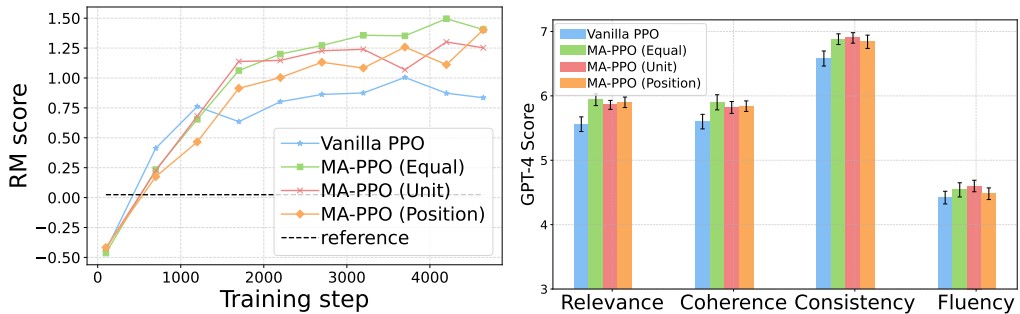

Figure 19: Performance of MA-PPO with different value function estimations in MA-PPO on TL;DR dataset for Gemma-2B model. **Left** test RM scores. **Right** GPT-4 scores on 4 dimensions.

## D.2 TERMINATION CONDITIONS ON DIFFERENT TASKS

In this section, we analysis the effectiveness of termination conditions on TL;DR, HH-RLHF, and APPS datasets. When implementing parsing-based termination condition on APPS dataset, we use a programming-language-based parser.[4] The results of TL;DR and HH-RLHF datasets are shown in Table 9 and Table 10. We can notice that parsing-based termination condition performs well on the HH-RLHF tasks, with higher RM score and win rate than fixed 5-gram based termination condition. While on the TL;DR dataset, parsing-based termination condition also achieves excellent performance compared to fixed 5-gram termination condition. On APPS dataset, parsing-based termination condition achieves the best results, except for the interview level task. These results demonstrate that construct macro action with linguistic information indeed brings performance gain to MA-PPO.

## D.3 IMPACT OF RLHF ON REWARD SCORE DISTRIBUTION

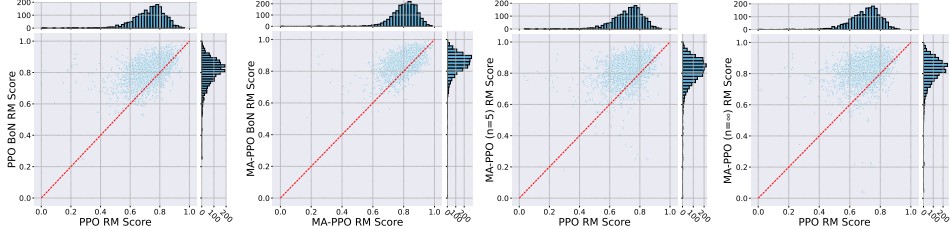

Figure 20: RM score shifting pattern after RLHF training. **Left** presents the RM score of best of 8 sampling on vanilla PPO compared to the vanilla PPO. **Mid Left** presents the RM score of best of 8 sampling on MA-PPO compared to the MA-PPO. **Mid Right** presents the RM score of MA-PPO ($n = 5$) compared to the vanilla PPO model. **Right** presents the RM scores of MA-PPO ($n = \infty$) compared to the vanilla PPO model.

We apply Best-of-$N$ sampling on both vanilla PPO and MA-PPO. The RM score shifting patterns for these methods are illustrated in Figure 20 (Left and Mid Left). From the results, we can conclude that Best-of-$N$ sampling continues to enhance the performance of RLHF models effectively.

In Figure 20 (Mid Right and Right), we compare the MA-PPO with vanilla PPO using settings of $n = 5$ and $n = \infty$, both of which demonstrate positive effects on the RM score distribution.

## D.4 IMPACT OF SAMPLING TEMPERATURE

In the previous experiments, the results were sampled with a temperature $temp = 0.8$ to align with the sampling strategy used during training. In this section, we examine the effect of sampling

---

[4]RedBaron https://github.com/PyCQA/redbaron

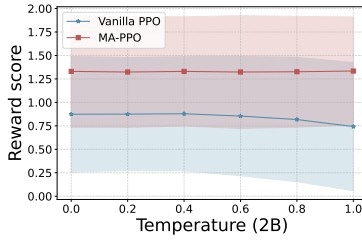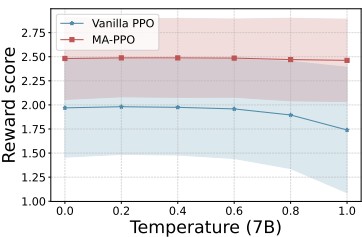

Figure 21: Test reward scores evaluated by the corresponding reward model for summarizations generated with different sampling temperature on the TL;DR dataset.

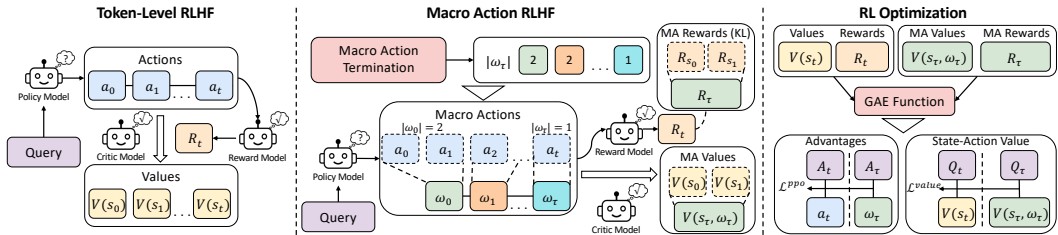

Figure 22: Illustration of the macro action-RLHF (MA-RLHF) framework.

temperature on response quality. We vary the temperature $temp \in \{0.0, 0.2, 0.4, 0.6, 0.8, 1.0\}$, and report the results in Figure 21. The performance of both methods remains stable when $temp < 0.8$. However, the performance of vanilla PPO begins to decline after $temp = 0.8$, whereas MA-PPO continues to demonstrate stable performance, even at $temp = 1.0$.

---

**Algorithm 1:** Framework of Macro Action RLHF.

---

**Input:** Prompts: $X = \{x_0, x_1, \ldots, x_n\}$; Policy model: $\pi_{\text{policy}}$; Reference model:
$\quad \pi_{\text{ref}}$; Critic model: $\pi_{\text{critic}}$; Reward model: $\pi_{\text{rm}}$; Termination rule $\zeta(\cdot)$ in
$\quad$ Section 3.2.1; Value function estimation $\sigma_{t_\tau}$ in Section D.1.
**Output:** Policy loss $\mathcal{L}^{\text{ppo}}$, Critic loss $\mathcal{L}^{\text{value}}$.

---

**foreach** *prompt $x_i$ in $X$* **do**
$\quad$ Make experience using policy model $y := \pi_{\text{policy}}(x)$;
$\quad$ Get value $V(s_t) := \pi_{\text{critic}}(x, s_t)$ at every time step $t \in [0, |y|)$;
$\quad$ Get reward score at current experience $r := \pi_{\text{rm}}(x, y)$;
$\quad$ Compute macro actions $\{\omega_\tau\}_{\tau=1}^m$ based on the termination rule $\{\omega_\tau\}_{\tau=1}^m := \zeta(y)$;
$\quad$ **foreach** *macro action $\omega_\tau$ in $\{\omega_\tau\}_{\tau=1}^m$* **do**
$\quad\quad$ Compute macro action value function
$$V^\pi(s_\tau, \omega_\tau) = \sum_{i=0}^{|\omega_\tau|} \sigma_{t_\tau+i} V^\pi(s_{t_\tau+i}, a_{t_\tau+i});$$
$\quad$ Obtain $\hat{A}_\tau$ and $\hat{Q}_\tau$ with GAE($V^\pi(s_\tau, \omega_\tau), r$);
$\quad$ Optimize $\mathcal{L}^{\text{ppo}} = \hat{\mathbb{E}} \left[ \min \left( \frac{\pi_\theta(\omega_\tau|s_\tau)}{\pi_{\theta_{\text{old}}}(\omega_\tau|s_\tau)} \hat{A}_\tau, \text{clip}(\frac{\pi_\theta(\omega_\tau|s_\tau)}{\pi_{\theta_{\text{old}}}(\omega_\tau|s_\tau)}, 1 - \epsilon, 1 + \epsilon)\hat{A}_\tau \right) \right]$
$\quad$ Optimize $\mathcal{L}^{\text{value}} = \hat{\mathbb{E}} \left[ \|V^\pi(s_\tau, \omega_\tau) - \hat{Q}_\tau\|^2 \right]$

---

# E  MA-RLHF ALGORITHMS

Figure 22 illustrates the framework of MA-RLHF. In practice, to implement MA-RLHF, once the macro actions are obtained via the termination function, we compute their value (as estimated by the critic model) and rewards (based on a per-token KL penalty) using the value function estimation. With these values and rewards, we apply Generalized Advantage Estimation (GAE) without modification to derive advantage estimates and state-action value functions. These advantage estimates and state-action value functions are then used to all tokens within the macro action during the opti-

mization of both the policy and critic models. The macro action RLHF algorithm, utilizing PPO, is detailed in Algorithm 1.

In this implementation, the introduced additional time complexity is in the option termination. While fixed $n$-gram based, randomized $n$-gram based, and perplexity based terminations achieves same time complexity, the time complexity of parsing based termination is related to the constituent tree which we applied DFS to obtain $|\omega_\tau|$. During the inference stage, our MA-PPO will not introduce additional complexity since it only works at the training stage.

We provide the Pytorch code for implementation of the macro action in PPO below:

**Obtain Macro Action Positions**

```python
def get_macro_action_positions(self, start, mask, termination='ngram',
    n_gram: int=None, ppl: List[torch.float16]=None, repeat_times: int=
    None, cutoff: int=None):
    sequence = [start]
    if termination == 'ngram':
        assert n_gram is not None
        current_count = 0
        for i in range(mask[:, start:].size(1) - 1):
            current_count += mask[0, start + i].item()
            if current_count == n_gram:
                sequence.append(start + i + 1)
                current_count = 0
    elif termination == 'randomized_ngram':
        k_list = torch.tensor([2, 3, 5, 10], dtype=int)
        k_list = torch.repeat_interleave(k_list, 3)
        k_list = k_list[torch.randperm(k_list.size()[-1])]
        indexed_k_list = torch.cumsum(k_list, dim=-1)

        sequence = [n for n in range(start, mask[:, start:].size(1) -
            1)]
        indexed_k_list = [x.item() for x in indexed_k_list if x.item()
            < len(sequence)]
        sequence = [start] + [sequence[i] for i in indexed_k_list]
    elif termination == 'ppl':
        assert ppl is not None
        for i in range(1, len(ppl)):
            if ppl[i] > ppl[i - 1]:
                sequence.append(start + i)
    elif termination == 'parser':
        if len(node.leaves()) < 1:
            return False, ma_length + 1
        if len(node.leaves()) < cutoff:
            sequence.append(ma_length + node.leaves())
            return True, ma_length + node.leaves()
        for nxt_node in node.childs():
            state, ma_length_ = dfs(nxt_node, ma_length)
            if !state:
                sequence[-1] = ma_length_
            ma_length = ma_length_
        return True, ma_length
    sequence.append(int(mask.size(1) - 1))
    return sequence
```

**Calculate Values / Rewards of Macro Action**

```python
def get_macro_action_values(self, values, mask, start, sequence):
    split_list = torch.diff(torch.tensor(sequence)).tolist()
    splited_values = torch.split(values[:, start:], split_list, dim=-1)
    splited_mask = torch.split(mask[:, start:], split_list, dim=-1)
    inplace_values = torch.zeros(1, len(split_list), dtype=values.dtype
        ).to(values.device)
    for idx, (value_i, mask_i) in enumerate(zip(splited_values,
        splited_mask)):
        masked_values = value_i[mask_i != 0]
        inplace_values[0, idx] = torch.mean(masked_values) if
            masked_values.numel() > 0 else 0.0
    return inplace_values
```

**Calculate Policy Model Loss**

```python
def policy_loss_macro_action(self, logprobs, old_logprobs, advantages,
    mask, sequence):
    log_ratio = (logprobs - old_logprobs) * mask
    ratio = torch.exp(log_ratio)

    # calculate loss with macro action
    split_list = torch.diff(torch.tensor(sequence)).tolist()
    split_ratio = torch.split(ratio, split_list, dim=-1)
    split_mask = torch.split(mask, split_list, dim=-1)

    pg_loss = 0.0
    total_mask_sum = 0.0
    for i in range(len(split_list)):
        ratio_i = split_ratio[i]
        mask_i = split_mask[i]
        advantages_i = advantages[:, i]

        pg_loss1 = -advantages_i * ratio_i
        pg_loss2 = -advantages_i * torch.clamp(ratio_i, 1.0 - self.
            cliprange, 1.0 + self.cliprange)
        pg_loss += torch.sum(torch.max(pg_loss1, pg_loss2) * mask_i)
        total_mask_sum += mask_i.sum()
    pg_loss = pg_loss / total_mask_sum
    return pg_loss
```

**Calculate Critic Model Loss**

```python
def critic_loss_macro_action(self, values, old_values, returns, mask,
    sequence):
    values_clipped = torch.clamp(
        values,
        old_values - self.cliprange_value,
        old_values + self.cliprange_value,
    )
    if self.compute_fp32_loss:
        values = values.float()
        values_clipped = values_clipped.float()

    # calculate loss with macro action
    split_list = torch.diff(torch.tensor(sequence)).tolist()
    splited_values = torch.split(values, split_list, dim=-1)
    splited_values_clipped = torch.split(values_clipped, split_list,
        dim=-1)
    splited_mask = torch.split(mask, split_list, dim=-1)

    total_vf_loss = 0.0
    total_mask_sum = 0.0

    for i in range(len(splited_values)):
        vf_loss1 = (splited_values[i] - returns[:, i])**2
        vf_loss2 = (splited_values_clipped[i] - returns[:, i])**2
        vf_loss = 0.5 * torch.sum(
            torch.max(vf_loss1, vf_loss2) * splited_mask[i])
        total_vf_loss += vf_loss
        total_mask_sum += splited_mask[i].sum()
    total_vf_loss = total_vf_loss / total_mask_sum
    return total_vf_loss
```

---

**PPO**

---

```
# In PPO algorithm
start = prompts.size()[-1] - 1
action_mask = attention_mask[:, 1:]
...
sequence = get_macro_action_positions(start, action_mask, termination='
    ngram', n_gram=n_gram)
macro_action_old_values = get_macro_action_values(old_values,
    action_mask, start, sequence)
macro_action_old_rewards = get_macro_action_values(old_rewards,
    action_mask, start, sequence)
advantages, returns = get_advantages_and_returns(sumed_old_values,
    sumed_old_rewards)

policy_loss = policy_loss_macro_action(policy_log_prob[:, start:],
    log_probs[:, start:], advantages, action_mask[:, start:], sequence)
critic_loss = critic_loss_macro_action(value[:, start:], old_values[:,
    start:], returns, action_mask[:, start:], sequence)
```

# F EVALUATION DETAILS

## F.1 GPT-4 EVALUATION PROMPTS

In our experiments, we take GPT-4 as a main judgment of the quality of policy models. The prompts used to generate win rates using GPT-4 are listed below. We utilize the `gpt-4o-05-13` for all of our experiments. The order of the responses generated by policy models is randomly chosen for all experiments.

---

**TL;DR GPT-4 Evaluation Prompt**

You will be given two summaries written for an article. Your task is to pick the better one between them, based on the four criteria. Please make sure you read and understand these instructions carefully.
**Relevance** - selection of important content from the source. The summary should include only important information from the source document. Annotators were instructed to penalize summaries which contained redundancies and excess information.
**Coherence** - the collective quality of all sentences. We align this dimension with the DUC quality question of structure and coherence whereby "the summary should be well-structured and well-organized. The summary should not just be a heap of related information, but should build from sentence to a coherent body of information about a topic."
**Consistency** - the factual alignment between the summary and the summarized source. A factually consistent summary contains only statements that are entailed by the source document. Annotators were also asked to penalize summaries that contained hallucinated facts.
**Fluency** - the quality of the summary in terms of grammar, spelling, punctuation, word choice, and sentence structure.
You should output single character to indicate which summary you think is better. 'A' stands for Summary A and 'B' stands for Summary B. If you think both summaries are equally good, output 'E'.
Article / Post: {article / post}
Summary A: {summary a}
Summary B: {summary b}
Your Choice (only a single character):

---

---

**HH-RLHF GPT-4 Evaluation Prompt**

For the following query to a chatbot assistant, which response is more helpful?
First provide a one-sentence comparison of the two responses and explain which you feel is more helpful. Second, on a new line, state only 'A' or 'B' to indicate which response is more helpful. If they are equally good or bad, state 'E'. Your response should use the json format, with "comparison" and "choice" as keys.
Query: {query}
Response A: {response a}
Response B: {response b}
Your Judgment:

---

**WebGPT Comparisons GPT-4 Evaluation Prompt**

You will be given two response written for an question. Your task is to pick the better one between them, based on these criteria.
**Factual accuracy** - which answer is more factually accurate?
**Coherence** - which answer is easier to follow?
**Usefulness overall** - all things considered, which answer would be more helpful to the person who asked this question?
You should output with a json format where the key is the criteria and the value is the choice you made, using 'A' stands for Response A and 'B' stands for Response B. If you think both responses are equally good, output 'E'.
Question: {question}
Answer A: {answer a}
Answer B: {answer b}
Your Judgment (you should also output the reason, note that you are allowed to think both responses are equally good, then output with 'E'):

---

## F.2 HUMAN EVALUATION

To estimate the quality from a human perspective, we collect human preference data on the TL;DR, HH-RLHF, and WebGPT datasets. Human annotators select the preferred response based on task-specific criteria. For TL;DR, the evaluation criteria focus on three main perspectives:

1. **Hallucination**: this considers whether the generated summary includes any additional information not present in the original post or article.
2. **Verbosity**: this assesses if the summary includes unnecessary context that could be removed without negatively impacting its quality.
3. **Overall Quality**: this measures the general coherence, informativeness, and readability of the generated summary.

For evaluation on TL;DR dataset, the annotators should first compare the overall quality of two responses. If overall qualities are equally good for responses, then they should choose the winner based on hallucination and verbosity.

In the context of HH-RLHF, annotators focus on the helpfulness of the responses:

1. **Instruction Following**: whether the generated response follows the requirements in the instruction
2. **Usefulness**: whether the advices in the response are applicable, and does the response ideally guide the user on what to do next.

Annotators are instructed to choose the response based on these aspects, while excluding superficial replies such as "You're welcome." For the WebGPT dataset, the primary evaluation factor is factual accuracy. Annotators are provided with retrieval information relevant to the question from the dataset to aid in their judgment. They are tasked with selecting the answer that most accurately matches the retrieved information.

During the evaluation process, annotators are presented with a prompt and two responses, each generated by either vanilla PPO or MA-PPO. To ensure impartiality and prevent annotators from

guessing which model produced which response, we shuffle the positions of the responses. Annotators are given three choices: response A wins, response B wins, or a tie. The results are then collected to calculate the win rates for each model.

For evaluations on the TL;DR and HH-RLHF datasets using 7B models, we conduct the human evaluation with 3 different annotators and collect their preference data to report the win rates. For all other human evaluations, we conduct them with a single annotator. The inter-rater agreement achieves an average of 68% on total 100 samples. On the TL;DR dataset the agreement is 64%, and on the HH-RLHF dataset the agreement is 72% across 50 samples per task.

# G  GENERATED EXAMPLES

## G.1  CASE STUDY

When evaluating the responses of MA-RLHF with human annotators, we observe that the MA-RLHF exhibits coherence and contextual appropriate abilities. We illustrate this phenomenon with an example by comparing MA-RLHF with the baseline in Table 11. We found that the MA-PPO method tends to generate responses with phrases more than the baseline method. Specifically, for nouns, it commonly includes adjectives for modification to make the generated summaries more accurate, such as "feeding indoor cat food".

## G.2  EXAMPLE RESPONSES

In this section, we demonstrate some examples of validation sets to highlight the superiority of MA-PPO. In Table 12, we feature examples from the TL;DR dataset. Compared to the responses generated by vanilla PPO, the responses from MA-PPO offer more concise and relevant details about the situation. Table 13 showcases a dialogue focusing on the differences between Spanish and Italian. MA-PPO provides a more helpful response, addressing both cultural and grammatical perspectives. In Table 14, we compare answers from the WebGPT dataset. The responses provided by MA-PPO are more factual and accurate compared to those generated by vanilla PPO.

Table 11: Example response from TL;DR dataset to illustrate the responses generated by MA-RLHF is more coherence and contextual appropriate compared to baseline. We highlight the context that we observed with green .

| Prompt | POST Subreddit: r/cats
Hello everyone! One of my cats is about 10 years old now, she is pretty much strictly indoors save for some time she spends on our screened in porch each day. (She likes to watch the birds in the yard while she suns herself by the pool, quite the princess). Anyway, when she was younger she was very active and quite small, however with age she has put on a pretty hefty amount of weight. I feed her indoor cat food for weight control, I've switched brands a few times trying to find something that works, I've cut back on feeding her by a lot (she gets very angry and demanding when she wants food but I don't give in) however, nothing really seems to work. I've tried cat toys, and bought a harness thinking I could try to walk her but she just lays down and looks at me like I'm stupid. Basically I just want to know if you all have any suggestions for exercise or food. I care about her and don't want this to get any worse. I also have another cat that eats the same amount and type of food as her and is a completely normal weight and only a year younger, however he is a male, not sure if that makes a difference in predisposition for weight gain. They are also both fixed. |
|---|---|
| Vanilla PPO | 10 year old cat has gained a noticeable amount of weight and is becoming increasingly inactive. Looking for suggestions for exercise or food to help her get healthier. Male and female cats are both affected. |
| Fixed 3-gram | 10 year old cat put on hefty amount of weight despite feeding indoor cat food for weight control. Any suggestions for exercise or food to help with weight control and prevent further weight gain? Specifically wondering if there are any suggestions for cats that are both female and fixed. |
| Fixed 5-gram | 10 year old indoor cat has put on significant amount of weight despite being fed low calorie/weight control food. Need suggestions for exercise or food to help with weight management. Also have another cat that eats same type/amount of food as her and is a completely normal weight and only a year younger, however he is a male. Any insights would be appreciated. |
| Fixed 10-gram | 10 year old cat is putting on significant weight despite feeding indoor cat food for weight control. Any suggestions for exercise or food to help her lose weight and keep her healthy? Also, is there a difference in predisposition for weight gain between male and female cats when fed same amount/type of food? |
| Fixed ∞-gram | One 10 year old cat has put on hefty amount of weight with age, feed indoor cat food for weight control, tried various brands and cut back feeding by lot, nothing really seems to work in terms of exercise or food, also have another cat same age and weight as her and completely normal weight and no issues currently however male and not sure if that makes difference predisposition for weight gain, both fixed. |
| Randomized n-gram | 10 year old cat has put on a lot of weight, feed her indoor cat food for weight control yet nothing seems to work. Any suggestions for exercise or food to address this? Other cat that eats same amount and type of food as her is a normal weight and a year younger, however is male, not sure if that makes a difference in predisposition for weight gain. |
| Perplexity based | 10 yo cat has put on significant amount of weight despite diet change and other attempts at exercise/activity. Any suggestions for exercise or food to help her lose weight? Also, any differences between males and females regarding predisposition for weight gain? |
| Parsing based | 10 year old indoor cat has gained significant amount of weight despite proper diet and exercise, potentially due to age related health issues. Need suggestions for exercise or food changes to help reduce weight and maintain proper body weight. Is weight gain common in older cats, and does gender play a factor in predisposition? |

Table 12: Example responses from TL;DR dataset. The summarization of MA-PPO exactly capture the details of the post.

| Prompt | POST Subreddit: r/running |
|---|---|
| | Last night I experienced my first ever absolutely miserable run where instead of feeling energized and accomplished when it was over, I lay on my bathroom floor shivering and hugging the toilet. |
| | I have had frustrating runs before and I have had runs when my knee/ankle/toe hurt and made me want to quit but last night was far worse than any of those. |
| | Earlier in the day, I ate a sandwich and fell asleep on the couch for a two hour surprise nap. When I woke up my stomach was slightly unsettled like I was about to have heartburn. I decided to do out for my long run anyways because I always have heartburn and it doesn't usually stop me. This run was scheduled to be my longest yet at 1:30:00 (I am training for a half). |
| | At about 45 minutes I started to get stomach cramps so I slowed how much water I was taking in. The entire run I had to walk several times which is unusual for me but my times were better than expected so I thought I was just pushing too hard and tried to take it easy on the way back. |
| | My post long run ritual is usually to drink several long drinks once I get back to the car since now I won't be jostling my innards around, I can do what I want. When I got home, I felt like I probably shouldn't try eating right away so I had about 1/3 bottle of Gatorade. |
| | Well...Gatorade was the wrong answer. Within 20 minutes I was violently heaving into my kitchen garbage and I lost all of the Gatorade and likely most of the water I had while running. |
| | My question is, do I blame my sandwich from earlier or my run? I am worried that 1:30:00 is my limit. I haven't considered quitting my half marathon training until now because everything up to this point has been fine. |
| **Vanilla PPO 2B** **RM Score:** 0.75 | 1:30:00 run was extremely stressful and painful. Do I blame sandwich from earlier or run itself? I am worried that 1:30:00 is my limit. |
| **MA-PPO 2B** **RM Score:** 1.55 | 1:30:00 long run resulted in severe stomach cramps and forced heavy heaving into kitchen garbage. Do I blame my sandwich from earlier or my run? Concerned that 1:30:00 is my limit. |
| **Vanilla PPO 7B** **RM Score:** 1.71 | 1:30:00 run caused serious stomach pains and I puked up all of the water I drank while running. Do I blame the sandwich for my stomach pain or the run itself? |
| **MA-PPO 7B** **RM Score:** 2.61 | 1:30:00 long run, had stomach cramps and had to walk often, drank Gatorade post run, vomited violently within 20 minutes, now worried that 1:30 is my limit, do I blame sandwich or run? Worried about quitting half marathon training. Help? |
| **Vanilla PPO 27B** **RM Score:** 3.45 | 45 minutes into my run my stomach started to cramp and I was vomiting profusely within 20 minutes of finishing my 1:30:00 long run. Do I blame my sandwich from earlier or my run? Does this mean my limit is 1:30:00 or is it something else? |
| **MA-PPO 27B** **RM Score:** 3.95 | I went out for a 1:30:00 long run yesterday and came home with stomach cramps and nausea. I drank 1/3 bottle of Gatorade when I got home and got violently sick. I lost that and most of the water I drank while running. Do I blame my sandwich from earlier or my run? I am worried that 1:30:00 is my limit and that I won't be able to finish my half marathon training. |

Table 13: Example responses from HH-RLHF dataset, MA-PPO method presents more helpful response compared to vanilla PPO, and is prefered by GPT-4 and human.

| | |
|---|---|
| **Prompt** | Human: Would you be able to explain the differences between the Spanish and Italian language? Assistant: Of course. Can you tell me more about the specific areas where you're interested in knowing more? Human: I'm thinking between the Spanish spoken in Mexico and Italian spoken in Italy. Assistant: |
| **Vanilla PPO 2B** **RM Score:** 1.07 | The main difference between Spanish and Italian is that Spanish is an Indo-European language with a very rich historical background, while Italian is an Italian Romance language derived from Latin. Spanish has influences from many different languages, while Italian has fewer and more limited influences from other languages. |
| **MA-PPO 2B** **RM Score:** 1.24 | Here are some of the main differences between Spanish spoken in Mexico and Italian spoken in Italy: 1. Spanish spoken in Mexico tends to be more relaxed and casual, while Italian spoken in Italy tends to be more formal and courteous. 2. Spanish in Mexico is heavily influenced by indigenous languages and has a rich vocabulary, while Italian is mostly derived from Latin and has a more simple and elegant sound. 3. Spanish is spoken throughout Latin America, while Italian is confined to the Italian peninsula. Overall, Spanish and Italian have many subtle differences that contribute to their unique cultural identities. |
| **Vanilla PPO 7B** **RM Score:** 0.80 | The main differences between the Spanish spoken in Mexico and the Italian spoken in Italy are in the pronunciation and grammar of the two languages. In Mexico, the Spanish spoken is much more similar to the Spanish spoken in Spain. In Italy, the Italian spoken is similar to the Italian spoken in other parts of Europe, but with its own unique style of pronunciation. In both languages, the grammar is very similar, though the Italian language does have some unique grammar rules. |
| **MA-PPO 7B** **RM Score:** 1.22 | Sure, here are some of the main differences between the Spanish spoken in Mexico and the Italian spoken in Italy: - Spanish is a latin language, which is a language that was derived from the latin language of ancient Rome. Italian is based on the ancient language of the romans of ancient Rome. - Spanish is spoken in a number of different countries throughout the world, whereas Italian is the native language of Italy, and is only spoken in some parts of northern Italy. - Spanish is spoken primarily in Latin America, and has taken on many local variations and differences due to the influence of local indigenous languages (i.e. Mexican Spanish is quite different from Spanish spoken in Argentina). Italian is primarily spoken in Italy and has more localized variations. - There are many differences in grammar, vocabulary, and pronunciation between Spanish and Italian that are reflected in the differences between Mexican Spanish and Italian. Some of the most important differences in grammar include the use of masculine and feminine gender to refer to objects, and the different use of the tenses. There are many differences in vocabulary that also exist between the two languages, and the pronunciation of each language also varies. |

Table 14: Example responses from WebGPT dataset.

| Prompt | What causes that sudden sinking, heavy, hot flush feeling when you become anxious and how are the effects felt so rapidly? |
|---|---|
| **Vanilla PPO 2B** **RM Score:** -0.68 | Anxiety has been linked to several physical symptoms, including a rapid heartbeat, trembling, dizziness, and a sense of impending doom. While some of these symptoms may be due to the anxiety-provoking situation causing the symptoms, some may be the result of an anxiety disorder itself. The symptoms that result from anxiety can be divided into physical symptoms, emotional symptoms, and behavioral symptoms. Physical symptoms of anxiety include nausea, dizziness, and a queasy or empty feeling in the stomach. Emotional symptoms include feeling helpless, hopeless, afraid, irritable, and worried. Behavioral symptoms include avoiding social interactions, remaining in bed all day, and not performing daily activities. |
| **MA-PPO 2B** **RM Score:** -0.41 | The "hot flush" sensation that some people experience when they become anxious is the result of the nervous system's fight-or-flight response. When the body is faced with a stressful or dangerous situation, the nervous system's adrenaline release causes blood to rush through the cardiovascular system to the muscles. This causes the body to become more efficient at using oxygen and energy, and it can also cause the skin to turn red and hot. This response is designed to help the body deal with stress and danger, and it can include a range of symptoms that impact different parts of the body. For example, elevated blood pressure can cause the kidneys to release more urine, and adrenaline can speed up the heart rate. It can also cause the mouth to dry out as the nervous system reacts to the perceived threat. In addition, anxiety can cause the chest to tighten and breathing to become quicker and shallower, which can cause the heart rate to increase. It can also cause the digestive system to slow down, and the muscles of the gastrointestinal tract to tighten, which can cause nausea and abdominal cramping. |
| **Vanilla PPO 7B** **RM Score:** -0.41 | There are several possible explanations as to why we feel hot and uncomfortable when anxious. One explanation is that anxiety causes the body to release hormones and chemicals into the bloodstream, which then circulate throughout the body and affect organs, tissues, and nerve cells. A different explanation is that a reaction to stress causes the body to release hormones and chemicals into the bloodstream, which then circulate throughout the body and affect organs, tissues, and nerve cells. Another explanation is that the body's stress response causes the release of hormones and chemicals into the bloodstream, which then circulate throughout the body and affect organs, tissues, and nerve cells. A fourth explanation is that anxiety triggers the release of cortisol, which increases blood sugar levels and causes the body to burn fat and muscle tissue for energy. |
| **MA-PPO 7B** **RM Score:** -0.39 | When we become anxious, it is because our brain has encountered something that it sees as a threat. The brain then sends out a warning signal, which takes the form of a fight-or-flight response. This response causes a number of physical effects, including increased heart rate and breathing rate, increased blood flow to muscles, and elevated blood sugar. This is because when we are threatened our body needs to be able to run or fight, and to do this we need increased oxygen and energy. Our body also needs to be able to protect itself, so we have increased blood flow to our muscles. When we are in a high-stress situation, our body produces serotonin and dopamine, which help us to feel good and focus on the task at hand. At the same time, our body produces norepinephrine and cortisol. Norepinephrine and cortisol work together to increase our heart rate, breath rate, and blood pressure, which in turn increases the flow of oxygen and nutrients to our muscles. Thus, when we are anxious, our body' way of protecting us is by increasing our awareness and increasing the flow of nutrients and oxygen to our muscles, so we can run or fight if needed. All of these processes occur so quickly because our body' systems are already set up to respond instantly to threats. |

