# OpenReview forum: "MA-RLHF: Reinforcement Learning from Human Feedback with Macro Actions"
_ICLR.cc/2025/Conference — ICLR 2025 Poster_

### Official Review · Reviewer_ziZM · 2024-10-23

**Soundness:** 2
**Presentation:** 3
**Contribution:** 2
**Rating:** 3
**Confidence:** 3

**Summary:**

The paper proposes the incorporation of options (in the RL sense) into the RL component of LLM fine tuning with RLHF, with the motivation being better temporal abstractions over generation of long token sequences. The method is evaluated on several benchmarks and model sizes and is shown to overall outperform RLHF by vanilla PPO.

**Strengths:**

The proposed method is (as mentioned by the authors) simple and could be of potential interest to practitioners.
There is also an extensive evaluation of the method both in terms of tasks/datasets and in terms of method (GPT-based and human annotators)

**Weaknesses:**

There are some significant conceptual weaknesses, as well as many technical issues.

Most importantly, there is a considerable gap between the stated/declared motivation and goals and what is actually being implemented.
The MA-RLHF algorithm only learns *one* option that is being applied everywhere -- there isn't a set of different options being learned, and there is no "macro" policy that chooses between them. As a result, generation really is driven by the (single) low-level policy alone, contrary to the authors statement that
- "Macro actions leverage meaningful linguistic structures" (Line 66; which is clearly false for several of the "termination" strategies later used such as n-grams!)
- "MA-RLHF operates at the macro action level, making decisions over sequences of tokens at a coarser temporal scale" (line 176-177).
- And so on.

What remains, effectively, is a particular way to chunk the rewards (and Advantage / Value estimates) that feed into the RL algorithm, in a way that might be different than the standard one. It could potentially be the case that this pushes the value estimation component more towards Monte-Carlo (rather than TD-learning / bootstrap) because Q (and A) are now fitted to a Bellman backup update that first sums over $\vert \omega_{\tau}\rvert$ observed rewards before bootstrapping (See $R_\tau$ definition; Line 218. This is also reflected in the authors analysis of Section 4.3.2, line 374, see more on that later). If this is the source of the performance gain, it can be implemented and evaluated much more directly, and the story about temporal abstractions is merely a distraction.

If the authors have a a good reason / explanation for this particular choice, it should be stated. And in any case, this limitation (or choice) must be clearly stated and acknowledged, preferably early on. Right now it's not even mentioned in the Limitations section (which is unjustifiably hidden in the appendix, Line 810).

Other than this main issue, there are plenty of technical issues, to a varying degree of severity. I expect some/most of them can be addressed in revisions. I have divided those into two sets, but I suspect these are not-exhaustive: the paper can definitely benefit from a closer proof-reading.

**"Moderate" Issues**
- The (Bengio et al. 2013) cite for the credit assignment problem (Abstract line 14, Intro Line 46) is out of place. That paper doesn't really deal with the RL credit assignment problem at all (although the term is mentioned, once, in the text).

- The comment "setting n → ∞ corresponds to the REINFORCE algorithm where the entire sequence is treated as a single macro action" (Line 399 and also Line 243) is completely unclear to me. REINFORCE most certainly **does not** treat the entire sequence as one action -- the policy is trained to generate standard "micro" actions at each time-step individually. The authors *might* mean to state something inline with the TD-based vs Monte-Carlo based method for estimating the Value/return, with PPO being an Actor-Critic variant and vanilla REINFORCE being a fully Monte-Carlo method. But this difference concerns the "critic" component rather than the actor. Moreover, the  (McGovern and Sutton 1998) cite for REINFORCE is wrong or at best confusing. (for REINFORCE the authors should cite Williams 92, or Sutton et al. 1999 if the claim is more generally on PG methods).

- What do we learn from Figure 3 that's not already shown in Figure 2?  There's nothing particularly interesting about the shape of the shown distributions besides the mean and variance, which are already presented in Fig. 2. If anything, other than showing the marginal histograms alone, there could be potential value in showing the correlation over individual examples and see if there's any structure in that (is the performance gain come from more easier/harder examples, or more uniform, etc). Also, if that figure is kept, it should be specified for which of the models and at which step at training the histograms are shown (judging by the scale I assume this is the 2B model at the end of training?).

- The authors claim that "MA-PPO achieves parity with vanilla PPO approximately 1.7 – 2 times faster during trainin" (Line 303). However the main difference in Figure 2 seems to be the final performance level which is higher for MA-PPO compared to PPO, and not the learning / convergence speed which is really comparable. This should either be made more quantitative / carefully evaluated, or less boldly stated.

- Figure 10: how come the marginal distribution of SFT RM scores changed between the first two panels and the last two panels?

**Minor issues**
There are, once again, many issues with phrasings, notation, etc.

Line 92: "An policy" -> "A policy"
Line 93: "The Markov Decision Processes" -> "A Markov decision process"
Line 98: The definition of Q(s,a) should include a conditioning on the fact that s_0 =s, a_0=a. It would also be better to write Q(s,a) rather than Q(s_t, a_t).
Line 98: The authors specify that reward is a function r(s,a) but then in defining Q use the formulation of r(s).

Line 108: The PPO discussion is rather dense. I feel that for a reader unfamiliar with the method, it will not be too helpful, and for familiar reader it will be redundant. Can be replaced by a reference to the appropriate paper, and shortened to a brief discussion of what Policy Gradient does (roughly, the inline Eq. in line 110), to be merged with the general RL formulation.

Line 111: Randomness of *the* initial state and so on.
Line 128: The acronyms for SFT and RM should be spelled out explicitly

Line 130: There's no use in writing r(x,y) if the reader don't know what x,y stand for. Better write "A reward model is trained to predict ...", then explain what the inputs are and what the goal is.

Line 140: Shouldn't the the equation read r(x,y) - beta KL? The way now written, the KL is not a penalty at all but a "bonus" to the reward (so higher KL will be encouraged)

Figure 1: I don't understand what we should read from the figure other than the fact that temporal abstractions are "a thing". And it is overall somewhat misleading, since it seems to imply that the optimized policy chooses macro actions (the MA-RLHF column, arrow from the MA-PPO network to the omega_tau action).

Line 145: Doesn't the state include the prompt as well? (other than the tokens already generated)

Line 236: The use of $r$ to denote the likelihood ratio or pi_theta/pi_theta_old is a really bad choice, because $r$ is reserved to denote the reward function. Is there actually a need to introduce more notation for the ratio? (note that in Equation 1, line 119, this notation is not being used).

Figure 2: Move the "(2B)" and "(7B)" to the ylabel ("RM Score (2B)"), or somewhere else -- otherwise it is confusing whether this 2B indicate scale of the time axis.

Figure 4: The fonts are tiny (in all figures really, but particularly here, only readable at 200% zoom or so).

**Questions:**

Somewhat more conceptually, the motivation about "essential linguistic structures and dependencies between adjacent tokens" (Line 54) is questionable. While the intuition is reasonable, in fact the very same argumentation can be made against the pre-training stage, where LLMs are trained to make single token predictions. But pre-trained LLMs already generate fluent/coherent output (surprisingly or not). It is a strong claim that making (one of) the fine-tuning step(s) more "global" can correct for the extensive pre-training. The power in RL methods might be the ability to design global reward/objectives (see also Ranzato et al. 2015, Choshen et al. 2019) which are better suited for aligning to designer preferences, but this is still true even if the model has to make only "micro" decisions. I think the paper can benefit from at least acknowledging these questions.

---

> ### Author Response · Authors · 2024-11-22
> **Response to Reviewer ziZM (Part 1/5)**
>
> Dear Reviewer ziZM,
>
> Thank you for your detailed review and thoughtful feedback. Below, we address each of your concerns in depth to clarify the key points and improve your understanding of our contributions.
>
> **Weaknesses**:
>
> > (W1): Most importantly, there is a considerable gap between the stated/declared motivation and goals and what is actually being implemented. The MA-RLHF algorithm only learns one option that is being applied everywhere -- there isn't a set of different options being learned, and there is no "macro" policy that chooses between them. As a result, generation really is driven by the (single) low-level policy alone, contrary to the authors statement that
> > * "Macro actions leverage meaningful linguistic structures" (Line 66; which is clearly false for several of the "termination" strategies later used such as n-grams!)
> > * "MA-RLHF operates at the macro action level, making decisions over sequences of tokens at a coarser temporal scale" (line 176-177).
> > * And so on.
>
> **Re #W1**: Thank you for your detailed feedback. We appreciate the opportunity to clarify the motivation and implementation of MA-RLHF.
>
> Our statement that "macro actions leverage meaningful linguistic structures" refers to the design of the termination strategies. While heuristic methods like $n$-gram chunking may not explicitly align with linguistic structures (but it can capture surface-form co-occurrence patterns between adjacent tokens), parsing-based and perplexity-based strategies are designed to incorporate meaningful linguistic patterns. For instance, parsing-based chunking explicitly leverages syntactic boundaries, which align with linguistic structures, while perplexity-based strategies approximate contextual coherence.
>
> We present the comparison of our method across different termination conditions below. Parsing-based methods, which explicitly leverage linguistic structures, achieve the highest performance in terms of GPT-4 win rates on the TL;DR and HH-RLHF datasets and yield the best results on APPS. Notably, simpler heuristics like $n$-gram-based termination also deliver competitive results, which demonstrates that macro actions, whether grounded in linguistic structures or surface-level co-occurrence patterns, can be of benefit for final performance and human evaluation results. The results are also listed in Appendix C.2.
>
> | **Dataset** | **Termination Condition** | **RM Score** | **GPT-4 Win Rate (v.s. PPO)** |
> |--|---------|--|--|
> | TL;DR       | Fixed 5-gram              | **1.40**     | **78%**                       |
> |             | Parsing                   | 1.37         | **78%**                       |
> |             | PPL                       | 1.27         | 72%                           |
> | HH-RLHF     | Fixed 5-gram              | 1.55         | 58%                           |
> |             | Parsing                   | **1.64**     | **62%**                       |
>
> | **Dataset** | **Termination Condition** | **Pass@$1$** |
> |--|--|--|
> | APPS        | Fixed 10-gram             | 5.45         |
> |             | Parsing                   | **5.56**     |
> |             | PPL                       | 5.26         |
>
> Regarding the claim that MA-RLHF operates at a coarser temporal scale, we emphasize that this is achieved by grouping contiguous token spans into macro actions. These spans are defined heuristically, such as by $n$-gram or parsing-based methods. A macro action over $|\omega_\tau|$ tokens is treated as an option in the MDP framework and is represented as:
> $$
> \pi_\theta (\omega_\tau | s_\tau) = \prod_{t=t_\tau}^{t_{\tau+1}} \pi_\theta (a_{t} | a_{<t}),
> $$
> preserving the autoregressive nature of LLMs, also refer to Section 3.2.2 Line#220. This formulation reduces the number of decision points during training, enabling better credit assignment across longer token spans, as supported by the RL literature [1, 2, 3].
>
> While our current implementation uses heuristic chunking methods, we recognize the computational challenges of learning a set of diverse macro actions from scratch in the language domain, where a sequence of $|\omega_\tau|$ tokens could have $\mathcal{V}^{|\omega_\tau|}$ combinations ($\mathcal{V}$ is the vocabulary size). By leveraging heuristics, we achieve effective temporal abstraction/chunking, which reduces the distance between the reward signal and decision points. This aligns with the overall goal of improving credit assignment and training efficiency.
>
> We hope these clarifications and additional results could fully address your concerns.

---

> ### Author Response · Authors · 2024-11-22
> **Response to Reviewer ziZM (Part 2/5)**
>
> > (W2): What remains, effectively, is a particular way to chunk the rewards (and Advantage / Value estimates) that feed into the RL algorithm, in a way that might be different than the standard one. It could potentially be the case that this pushes the value estimation component more towards Monte-Carlo (rather than TD-learning / bootstrap) because Q (and A) are now fitted to a Bellman backup update that first sums over |ωτ| observed rewards before bootstrapping (See Rτ definition; Line 218. This is also reflected in the authors analysis of Section 4.3.2, line 374, see more on that later). If this is the source of the performance gain, it can be implemented and evaluated much more directly, and the story about temporal abstractions is merely a distraction.
>
> > If the authors have a a good reason / explanation for this particular choice, it should be stated. And in any case, this limitation (or choice) must be clearly stated and acknowledged, preferably early on. Right now it's not even mentioned in the Limitations section (which is unjustifiably hidden in the appendix, Line 810).
>
> **Re #W2**: Thank you for pointing this out. Our primary goal is to address the credit assignment problem by reducing the temporal resolution of decision points through macro actions [1]. By summing rewards over macro actions, we effectively reduce the variance in advantage estimation, as the reward is more closely tied to the decision context. This aligns with the Monte-Carlo perspective, which aggregates feedback over extended temporal spans.
>
> We acknowledge that this choice shifts the balance between Monte-Carlo and TD-based updates. We have updated the limitations section to ensure transparency about these trade-offs.
>
> **"Moderate" Issues**:
>
> > (W3): The (Bengio et al. 2013) cite for the credit assignment problem (Abstract line 14, Intro Line 46) is out of place. That paper doesn't really deal with the RL credit assignment problem at all (although the term is mentioned, once, in the text).
>
> **Re #W3**: We appreciate the reviewer’s feedback regarding the citation of (Bengio et al. 2013). We agree that this reference does not specifically address RL credit assignment and have replaced it with more appropriate works from the RL literature [1, 2, 3].
> Macro actions (or options) have been widely studied as a way to mitigate the credit assignment problem by reducing the temporal resolution of decision-making and associating rewards with extended actions [1, 2, 3]. As noted in [1], macro actions encapsulate sequences of adjacent decisions, providing temporal abstraction that facilitates long-term credit assignment. By chunking trajectories into coarser units, this method addresses challenges related to sparse and delayed rewards.
>
> In our method, macro actions correspond to token spans that reduce the gap between reward signals (provided at the sequence level) and decision points, thereby stabilizing learning and improving efficiency. These benefits are aligned with findings in [1, 2, 3], which highlight the utility of temporal abstraction in RL.
>
> [1] Pignatelli, E., Ferret, J., Geist, M., Mesnard, T., van Hasselt, H., Pietquin, O., & Toni, L. (2023). A survey of temporal credit assignment in deep reinforcement learning. arXiv preprint arXiv:2312.01072.
>
> [2] Machado, M. C., Barreto, A., Precup, D., & Bowling, M. (2023). Temporal abstraction in reinforcement learning with the successor representation. Journal of Machine Learning Research, 24(80), 1-69.
>
> [3] Pang, Z. J., Liu, R. Z., Meng, Z. Y., Zhang, Y., Yu, Y., & Lu, T. (2019, July). On reinforcement learning for full-length game of starcraft. In Proceedings of the AAAI Conference on Artificial Intelligence (Vol. 33, No. 01, pp. 4691-4698).

---

> ### Author Response · Authors · 2024-11-22
> **Response to Reviewer ziZM (Part 3/5)**
>
> > (W4): The comment "setting n → ∞ corresponds to the REINFORCE algorithm where the entire sequence is treated as a single macro action" (Line 399 and also Line 243) is completely unclear to me. REINFORCE most certainly does not treat the entire sequence as one action -- the policy is trained to generate standard "micro" actions at each time-step individually. The authors might mean to state something inline with the TD-based vs Monte-Carlo based method for estimating the Value/return, with PPO being an Actor-Critic variant and vanilla REINFORCE being a fully Monte-Carlo method. But this difference concerns the "critic" component rather than the actor. Moreover, the (McGovern and Sutton 1998) cite for REINFORCE is wrong or at best confusing. (for REINFORCE the authors should cite Williams 92, or Sutton et al. 1999 if the claim is more generally on PG methods).
>
> **Re #W4**: Thank you for your comments. To clarify, we did not intend to claim that REINFORCE treats the entire sequence as one action in its standard formulation. Instead, our reference to REINFORCE as modeling the "entire generation as a single macro action" draws from [1], which discusses how REINFORCE can be conceptualized as a bandit problem when viewed through the lens of sequence-level optimization. We refer the reviewer to **Section 3.3 in [4]** for the discussion of this viewpoint.  Specifically, when modeling an entire sequence as a macro action in our method, the critic becomes irrelevant, and the policy is optimized solely based on rewards provided at the sequence level, similar to the Monte Carlo nature of REINFORCE.
>
> Our approach uses macro actions to reduce the resolution of decision points, aligning with Monte Carlo-style updates where the temporal scale spans multiple tokens or even the entire sequence. We agree that this distinction primarily pertains to the critic component rather than the actor and have revised the manuscript to ensure this explanation is more precise and consistent.
>
> We hope this clarification resolves any misunderstandings.
>
> Also, we have replaced it with more appropriate citations for REINFORCE, including [5][6].
>
> [4] Ahmadian, A., Cremer, C., Gallé, M., Fadaee, M., Kreutzer, J., Pietquin, O., ... & Hooker, S. (2024). Back to basics: Revisiting reinforce style optimization for learning from human feedback in llms. arXiv preprint arXiv:2402.14740. https://arxiv.org/pdf/2402.14740
>
> [5] Ronald J Williams. Simple statistical gradient-following algorithms for connectionist reinforcement learning. Machine learning, 8:229–256, 1992. https://link.springer.com/article/10.1007/bf00992696
>
> [6] Richard S Sutton, David McAllester, Satinder Singh, and Yishay Mansour. Policy gradient methods for reinforcement learning with function approximation. Advances in neural information processing systems, 12, 1999. https://papers.nips.cc/paper_files/paper/1999/hash/464d828b85b0bed98e80ade0a5c43b0f-Abstract.html
>
> > W5: What do we learn from Figure 3 that's not already shown in Figure 2? There's nothing particularly interesting about the shape of the shown distributions besides the mean and variance, which are already presented in Fig. 2. If anything, other than showing the marginal histograms alone, there could be potential value in showing the correlation over individual examples and see if there's any structure in that (is the performance gain come from more easier/harder examples, or more uniform, etc). Also, if that figure is kept, it should be specified for which of the models and at which step at training the histograms are shown (judging by the scale I assume this is the 2B model at the end of training?).
>
> **Re #W5**: Figure 3 provides additional insight into the reward distribution for each method. While Figure 2 summarizes mean performance, Figure 3 shows the distributional changes. We have updated the caption and main text to highlight these points explicitly. Additionally, we now specify that the histograms correspond to the 2B model at the end of training (4.6k steps).
>
> While pinpointing the exact origin of performance gains across the distribution is inherently complex—given the challenges of interpreting the optimization process of LLMs—we provide qualitative examples to show how sequences are grouped into macro actions under various termination conditions. These examples, separated by @@, are included in our response to Reviewer 85YK Question 2 and demonstrate the practical impact of different macro action chunking strategies. The examples are shown in the next response, i.e., Part 4/5.

---

> ### Author Response · Authors · 2024-11-22
> **Response to Reviewer ziZM (Part 4/5)**
>
> **Re #W5**:
>
> **Example from TL;DR dataset:**
>
> | Termination                    | Macro Actions                                                                                                                                                                                    |
> |--------------------------------|--------------------------------------------------------------------------------------------------------------------------------------------------------------------------------------------------|
> | (Original Response)            | 20M, moved to Europe, met some people but they quickly disappeared, want to celebrate, but I'm still alone, would it seem weird or desperate if I go out alone?                                  |
> | Fixed 5-gram based termination | 20M, moved@@ to Europe, met some@@ people but they quickly disappeared@@, want to celebrate,@@ but I'm still@@ alone, would it seems@@ weird or desperate if I@@ go out alone?@@                 |
> | Perplexity-based termination   | 20M, moved to@@ Europe@@,@@ met some people@@ but they@@ quickly disappeared@@, want@@ to celebrate,@@ but I'm still@@ alone@@,@@ would it seems@@ weird or desperate if@@ I go out@@ alone@@?@@ |
> | Parsing-based termination      | 20M,@@ moved to Europe,@@ met some people but they quickly disappeared,@@ want to celebrate, but@@ I'm still alone, would it seems@@ weird or desperate if@@ I go out alone?@@                   |
>
> **Example from APPS dataset**:
>
> **(Original Response)**:
> ```python
> def queue_time(customers, n):
>     tills = [0] * n
>     for i in range(len(customers)):
>         tills[i % n] += customers[i]
>     return max(tills)
> ```
> **Fixed 3-gram based termination**:
> ```python
> def queue_@@time(customers@@, n):@@
>     till@@s = [@@0] *@@ n
>     @@for i in@@ range(len@@(customers)):@@
>         till@@s[i@@ % n]@@ += customers[@@i]
> @@    return max@@(tills@@)@@
> ```
> **Fixed 10-gram based termination**:
> ```python
> def queue_time(customers, n):
> @@    tills = [0] * n
> @@    for i in range(len(customers)):@@
>         tills[i % n] +=@@ customers[i]
>     return max(till@@s)@@
> ```
> **Perplexity-based termination**:
> ```python
> def queue@@_@@time@@(customers@@, n@@):
> @@    tills@@ = [@@0]@@ *@@ n
> @@    @@for i@@ in range@@(len@@(customers@@)):
> @@        tills@@[@@i % n@@] +=@@ customers[@@i@@]@@
>     return max@@(till@@s@@)@@
> ```
> **Parsing-based termination**:
> ```python
> def queue_time(customers, n):@@
>     @@tills = [0] * n@@
>     @@for i in range(len(customers)):@@
>         @@tills[i % n] += customers[i]@@
>     @@return max(tills)@@
> ```
>
> We hope these clarifications and additions address your concerns, and we look forward to further engagement on this topic.
>
> > (W6): The authors claim that "MA-PPO achieves parity with vanilla PPO approximately 1.7 – 2 times faster during training" (Line 303). However the main difference in Figure 2 seems to be the final performance level which is higher for MA-PPO compared to PPO, and not the learning / convergence speed which is really comparable. This should either be made more quantitative / carefully evaluated, or less boldly stated.
>
> **Re #W6**: Thank you for highlighting this point. We apologize for the lack of clarity in our original statement. To clarify, our claim pertains to the learning efficiency of MA-PPO compared to vanilla PPO, specifically in achieving comparable performance levels. Quantitatively, MA-PPO reaches the same reward score as PPO approximately 1.7× faster (e.g., 1.7k vs. 3.7k steps for the 2B model), which demonstrates its improved learning efficiency rather than faster overall convergence.
>
> We have revised the manuscript to make this distinction explicit and provide quantitative details to substantiate the claim. We hope this addresses your concern, and we appreciate your feedback in helping us refine the presentation of our results.
>
> > (W7): Figure 10: how come the marginal distribution of SFT RM scores changed between the first two panels and the last two panels?
>
> **Re #W7**: Thank you for pointing out this! The x-axis labels in the last two panels were incorrect and have been fixed to reflect SFT RM scores. We have replaced the panels with updated figures to ensure consistency.
>
> > (W8): Minor issues There are, once again, many issues with phrasings, notation, etc.
>
> **Re #W8**: Thank you for pointing out these issues. We have addressed all the specific concerns (e.g., Line 92, Figure 4 font size) and performed a thorough proofread to improve clarity, notation, and overall presentation.

---

> ### Author Response · Authors · 2024-11-22
> **Response to Reviewer ziZM (Part 5/5)**
>
> **Questions**:
>
> > (Q1): Somewhat more conceptually, the motivation about "essential linguistic structures and dependencies between adjacent tokens" (Line 54) is questionable. While the intuition is reasonable, in fact the very same argumentation can be made against the pre-training stage, where LLMs are trained to make single token predictions. But pre-trained LLMs already generate fluent/coherent output (surprisingly or not). It is a strong claim that making (one of) the fine-tuning step(s) more "global" can correct for the extensive pre-training. The power in RL methods might be the ability to design global reward/objectives (see also Ranzato et al. 2015, Choshen et al. 2019) which are better suited for aligning to designer preferences, but this is still true even if the model has to make only "micro" decisions. I think the paper can benefit from at least acknowledging these questions.
>
> **Re #Q1**: Thank you for this insightful question. While pre-training indeed achieves coherence through token-level objectives, our method targets specific alignment challenges inherent in RLHF. By grouping adjacent tokens into macro actions (e.g., $n$-grams or parsing-based spans), MA-RLHF enables coarser temporal abstraction and improves credit assignment across longer token sequences [1,2,3]. This design mitigates the credit assignment problem by reducing the decision points, aligning RLHF more effectively with sequence-level objectives.
>
> To address potential ambiguities, we have revised the manuscript to replace "linguistic structures" with **"local co-occurrence patterns in adjacent tokens"**, which better reflects the heuristic nature of our chunking strategies. Specifically, macro actions group adjacent tokens using various heuristic methods, such as $n$-gram-based, parsing-based, and perplexity (PPL)-based strategies, each with a distinct relationship to linguistic structures.
>
> 1. $n$-gram-based: This approach groups a fixed number of adjacent tokens, capturing local patterns and co-occurrence relationships between neighboring tokens. While $n$-grams do not directly model linguistic structures like syntax or semantics, they reflect the proximity and frequency patterns often present in natural language. For example, common phrases or repetitive structures can be well-represented by $n$-grams.
>
> 2. Parsing-based: This method directly leverages linguistic structures by grouping tokens based on syntactic dependencies or constituent phrases. For example, phrases such as "the quick brown fox" may be chunked as a single macro action based on grammatical boundaries, aligning more closely with linguistic hierarchies.
>
> 3. PPL-based: Perplexity-based termination groups tokens dynamically, based on their probability under the language model. This method does not explicitly rely on linguistic structures but instead reflects the model's uncertainty, potentially grouping tokens that form coherent or predictable units in the text.
>
> In summary, while $n$-gram-based strategies focus on local co-occurrence patterns, parsing-based methods explicitly integrate linguistic structures, and PPL-based methods adaptively reflect fluency and coherence as perceived by the model. By applying these macro action strategies, MA-RLHF can flexibly leverage heuristics and structure, which enables robust alignment with sequence-level objectives and improves credit assignment.
>
> Additionally, during inference, MA-RLHF introduces no extra computational cost, as the generation process remains token-based and fully compatible with standard autoregressive models. This ensures both practicality and efficiency in real-world deployment.
> We also appreciate your suggestion regarding global reward/objectives. While this perspective provides an alternative lens to understand the effectiveness of RL methods, our choice of macro actions aligns well with established frameworks in RL literature, emphasizing sequence chunking, grouping, and decision-resolution reduction. We hope this addresses your concerns, and we look forward to any further engagement.
>
> [1] Pignatelli et al., (2023). A survey of temporal credit assignment in deep reinforcement learning. arXiv preprint arXiv:2312.01072.
>
> [2] Machado et al., (2023). Temporal abstraction in reinforcement learning with the successor representation. Journal of Machine Learning Research, 24(80), 1-69.
>
> [3] Pang et al., (2019). On reinforcement learning for full-length game of starcraft. In Proceedings of the AAAI Conference on Artificial Intelligence (Vol. 33, No. 01, pp. 4691-4698).
>
> ---
> We sincerely thank you for your detailed review and valuable feedback, which have helped us significantly improve the clarity and rigor of our submission. We hope that the additional explanations and updates address your concerns, and we kindly request you to reconsider your score in light of these clarifications.

---

> ### Comment · Reviewer_ziZM · 2024-11-25
>
> The authors response has addressed many of the minor/technical issues, and to reflect that I have updated the "presentation" score to 3.
> Nevertheless, the more fundamental issues persist -- there is simply no "macro actions" by any meaningful sense of that term in this work. There definitely isn't a "reducing the temporal resolution of decision points through macro actions", because the way decisions are made are not affected at all compared to "vanilla" PPO. The actual method is doing something entirely different -- something which might still be beneficial on its own, but shouldn't be presented as something it isn't (once again, the very small differences in performance from the different termination strategies are a demonstration of that).

---

> ### Author Response · Authors · 2024-11-25
> **Response to Reviewer ziZM Follow-Up**
>
> Dear Reviewer ziZM,
>
> Thank you for engaging with our work. As noted in our **earlier response to #W1**, **directly enumerating all possible "macro actions" in language space is computationally infeasible**. For a vocabulary size of 65,536 and a macro action length of  $n = 5$, the action space would balloon to $1.2×10^{24}$ possible combinations. To manage this, we use a factorization approach consistent with the autoregressive nature of GPT models. Specifically:
>
> 1. **Definition of Macro Actions**: We define macro actions as sequences of adjacent tokens (e.g., $n$-grams, parsing-based spans, or perplexity-based spans). These sequences are treated as cohesive units for reward assignment and advantage computation during training. While the tokens within a macro action are **generated sequentially (due to the autoregressive model)**, the reward and advantage are computed for the entire macro action, effectively operating at a coarser temporal resolution.
> 2. **Distinction from Micro Actions**: Generating tokens one-by-one does not imply decision-making at the micro level. The macro action is defined over contiguous tokens, and our training process optimizes token transitions and their aggregated rewards as a single unit. This distinction ensures that the decision-making process operates at the macro level during training. Specifically, we define the macro action as:
> $$
> \pi_\theta (\omega_t | s_t) = \prod_{j=0}^{|\omega_t|-1} \pi_\theta (a_{t+j} | a_t, a_{t+1}, \cdots, a_{t+j}),
> $$
> to **preserve the autoregressive nature of LLMs**. Treating an entire sequence of contiguous tokens (e.g., 5 tokens) as a single classification choice (option) would be **computationally prohibitive and necessitate fundamental changes to the underlying LLM framework**. This factorized representation allows us to model macro actions while maintaining compatibility with standard LLM architectures.
>
> 3. **Impact on Training Dynamics**: Compared to vanilla PPO, MA-PPO reduces the temporal distance between decision points and reward signals by grouping tokens into macro actions. This grouping improves credit assignment efficiency, as evidenced by faster learning dynamics and higher reward scores in our experiments. *We respectfully ask for clarification on the statement, "the way decisions are made are not affected at all compared to vanilla PPO"*. If this refers to the token-by-token generation process, we emphasize that our framework focuses on macro-level optimization during training while maintaining token-level inference to ensure compatibility with autoregressive LLMs.  **For inference, generating $n$ tokens at a time as a macro action is functionally equivalent to generating tokens one by one until reaching the EOS token**.
>
> We would also like to highlight that our detailed responses addressed the other comments (#W3–#W8). Specifically:
> * **Revised Presentation**: We have clarified improved notation, corrected figure captions, and addressed all technical inconsistencies, as outlined in our previous responses.
> * **Acknowledgment of Limitations**: We explicitly acknowledged the heuristic nature of our macro action definitions and the differences between our approach and classical macro action frameworks, to ensure clarification.
>
> We sincerely hope that these responses, along with the additional experiments and manuscript revisions, have clarified any misunderstandings and demonstrated the contributions and robustness of our work.
>
> Additionally, **we kindly ask whether our responses have adequately addressed your concerns regarding W3–W8**. If they have, **we respectfully request you to reconsider your assessment** of our submission. If there are any remaining questions or concerns, **we would be more than happy to engage further in addressing them**.
>
> Thank you again for your thoughtful feedback! Looking forward to your further comments.

---

> ### Author Response · Authors · 2024-11-26
> **Request for Re-review and Discussion**
>
> Thank you again for your feedback. As the response period nears its end very soon, we kindly ask if our responses and revisions have addressed your concerns. If so, we respectfully request you to reconsider your assessment. Please let us know if there are any remaining points to discuss.

---

> ### Comment · Reviewer_ziZM · 2024-11-26
>
> "We define macro actions as sequences of adjacent tokens": this is clearly inconsistent with the definition actually used in the paper, lines 158-161. If we are to understand (as the paper wants us to) macro-actions as *options* (in the RL sense), then they have much more structure associated with them:
>
> Crucially, macro-actions make sense when **different macro-actions correspond to different "low-level"/micro policies**. And so, while generation is still sequential/autoregressive, the agent has, in fact, some control on a coarser temporal scale by choosing *the policy from which* it is going to sample the next K actions. The case here is that there is only 1 macro-action, that is (necessarily) always chosen at each "macro" decision point, completely nullifying the meaning of these macro-actions. The "termination condition" alone is also somewhat empty, because it doesn't matter if one macro-action terminates or not, the next macro-action is going to keep generating from the very same micro-action policy.
>
> Once again, the point is not about the autoregressive nature of sampling/generation *per se* (in the Options framework, choosing actions for the environment is also sequential, in fact it has to be so because the next-state is not even determined, in the general case, before current action is communicated to the environment).
>
> So, as noted previously, the entire method seems to boil down to some tweaking of the value/advantage estimate in the sense that it emphasizes more Monte-Carlo estimation over Bootstrap/TD. This might be of some benefit for itself, as the results suggest. But the entire motivation, introduction, and interpretation in the paper is written as if the method is actually about macro-actions, temporal abstraction, and so on. The paper should describe what is actually being done, and not mislead readers into understanding something completely different.
>
> Regarding the other comments from my review:
> Yes, these have been largely addressed, and I thank the authors for their detailed response to all of them. I have reflected this in my updated "presentation" assessment. As mentioned in my initial review, I expected that most of these could in fact be addressed, and these were not the key issues underlying my evaluation of the paper.

---

> ### Author Response · Authors · 2024-11-28
>
> Thank you for your thoughtful follow-up comments! In Line#159, the "action selection among macro actions" could be a typo, which has been revised as "action" ($\pi: \mathcal{S} \times \mathcal{A} \rightarrow [0,1]$). To clarify:
>
> 1. **Clarification on macro actions** in LLM context: As noted in the RL literature [1, 2], "macro-actions are policies with termination conditions". At each time step, the agent takes either a primitive action or a macro action until reaching the termination condition, encompassing a sequence of low-level actions. **Setting the "actions can be considered a special case of options"  (macro actions), namely "single-step" or primitive options (macro actions)** [1]. It is worth clarifying that the decision point of both previous RL literature and our work is NOT on a whole span and nullifying the termination condition. Instead, the continous action/token subsequence until the termination condition are defined as the "macro actions".
>
> 2. In the specific context of LLMs, **defining new macro actions as one options (e.g., one macro action per $n$-gram) would require re-architecting the LLM’s vocabulary and retraining the model**, which is computationally infeasible. Instead, we preserve the autoregressive nature of LLMs, where token probabilities are sequentially factorized, allowing our method to efficiently abstract token-level decisions into macro actions without modifying the LLM architecture.
>
> 3. The relationship between macro actions and primitive tokens in our method can be interpreted through the lens of token merging in BPE algorithms. From the pair-merging nature of BPE algorithm, we can treat each merged token in the vocabulary as "pseudo macro actions" on top of primitive actions (constituent tokens before merging). Applying new macro actions is equivalent to expanding the BPE vocabulary entries, which can be treated as another view for interpreting the relations between macro actions (merged tokens) and primitive tokens (i.e., the primitive tokens that cannot be split). This also corresponds to our previous motivation on "de-tokenization" in the introduction of our paper.
>
> Thank you again for your constructive feedback! We hope this could clarify the implementation of macro actions and have made revisions in the updated manuscript (Line#159, Line#218-222, and Line#543-546). If there are any remaining concerns, we would be happy to discuss further.
>
> **References**:
>
> [1] Sutton, Richard S., Doina Precup, and Satinder Singh. "Between MDPs and semi-MDPs: A framework for temporal abstraction in reinforcement learning." Artificial intelligence 112.1-2 (1999): 181-211.
>
> [2] McGovern, Amy, and Richard S. Sutton. "Macro-actions in reinforcement learning: An empirical analysis." Computer Science Department Faculty Publication Series (1998): 15.

---

> ### Author Response · Authors · 2024-12-02
> **Request for Further Discussion**
>
> Dear Reviewer ziZM,
>
> Thank you for your insightful comments and helpful feedback! We hope our detailed response have addressed your concerns. **As the extended deadline for discussion period will conclude very soon, we respectfully request your further engagement and would be happy to address any remaining comments!**

---

### Official Review · Reviewer_QYzk · 2024-10-28

**Soundness:** 3
**Presentation:** 3
**Contribution:** 3
**Rating:** 6
**Confidence:** 3

**Summary:**

The main idea is to use a larger chunk of text as the action in RLHF, instead of just one token (e.g., one word or one sub-word unit). What’s a larger chunk? The authors have tried multiple strategies: n-gram-based, parsing-based, perplexity-based “termination” of macro actions (i.e., segmenting tokens into a macro-action) – those are simple heuristics-based strategies introduced on page 4. Algorithms are built on top of naive PPO.

Experiments are done using Gemma-2B, Gemma-7B, and Gemma-2-27B. Results on multiple tasks (summarization, dialogue or instruction following including HH-RLHF and WebGPT, code generation) show improvements.

**Strengths:**

The ideas are intuitive. Recent progress has shown that RL is promising in general so it’s great that the authors are investigating this direction.

Multiple macro action termination strategies are explored. Experiments on code generation / program synthesis is helpful.

Interesting analysis on rejection sampling vs. temperature.

Helpful figures & illustrations in the appendix.

**Weaknesses:**

Results are great and surprising (not weaknesses); e.g., in Figure 4, the newly trained models are strongly preferred over the baseline (vanilla PPO). But:
- What’s the main reason here? Is there a comprehensive analysis that makes sure that the win rate is not based on simple features like length?
- Are baselines trained with a sufficient number of steps?


The equation for R is incorrect in Section 2.2.
- KL is between two distributions, but the authors wrote that the KL is between two real numbers.
- The last term should be -KL( \pi_\theta ( \cdot | x) || \pi_\text{ref} (\cdot | x)); in particular, KL(p || q) does not equal to -KL(q || p). Currently if you expand the KL, the interpretation is that if \pi_sft (y|x) is large, then we should make \pi_\theta (y|x) really small, in order to maximize R.

Other issues in Section 2
- Can gamma be equal to 1? (Section 2.1)
- In Eq. (1) it’s unclear if A_t is computed using \theta or \theta_old – it’s never defined; similarly, how about Section 3 line 233 – what model would the authors use to compute the advantage?
- In Eq. (1) what’s the expectation over?
- At the end of page 2, the indices start from 0 (e.g., s_0, a_0); but on page 3 line 145, the indices start from 1

**Questions:**

A natural way is based on BPE-like automatic construction of a dictionary with most frequent n-grams. Have the authors considered this strategy?

Do the authors think that leave-one-out REINFORCE or other recent PPO variants would also benefit from the authors’ algorithm (macro-actions)?

---

> ### Author Response · Authors · 2024-11-22
> **Response to Reviewer QYzk (Part 1/2)**
>
> Dear Reviewer QYzk,
>
> Thank you for your thorough review and valuable comments. We address your concerns and questions below:
>
> **Response to Weakness**:
>
> > (W1): Results are great and surprising; e.g., in Figure 4, the newly trained models are strongly preferred over the baseline (vanilla PPO).
> > * What’s the main reason here? Is there a comprehensive analysis that makes sure that the win rate is not based on simple features like length?
> > * Are baselines trained with a sufficient number of steps?
>
> **Re #W1**: We attribute the strong performance of MA-PPO to its ability to aggregate actions into macro-actions, which improves value function estimation, reduces decision points, and stabilizes policy updates.
>
> To ensure performance gains are not due to length bias, we conducted both human and GPT-4 evaluations on the TL;DR and HH-RLHF datasets, observing high agreement rates between them (61% average). Additionally, we applied a debiasing approach using a regression model inspired by Length-Controlled AlpacaEval [1], even though reproducing their full methodology is infeasible due to our limited sample size (50 samples per task).
>
> We trained an XGBoost regression model to predict reward scores based on response lengths. Using 7,500 training instances (combined data from vanilla PPO, MA-PPO, and dataset references) and 250 testing instances per method, we calculated debiased rewards by subtracting predicted rewards from the actual scores. Results are as follows:
>
> | **Size** | **Model** | **RM Score (TL;DR)** | **Debiased RM Score (TL;DR)** | **RM Score (HH-RLHF)** | **Debiased RM Score (HH-RLHF)** |
> |----------|-----------|----------------------|-------------------------------|------------------------|---------------------------------|
> | 2B       | PPO       | 0.86                 | 0.39                          | 1.33                   | 0.70                            |
> | 2B       | MA-PPO    | **1.39**             | **0.53**                      | **1.53**               | **0.90**                        |
> | 7B       | PPO       | 1.91                 | 0.40                          | 1.07                   | 0.52                            |
> | 7B       | MA-PPO    | **2.50**             | **0.58**                      | **1.24**               | **0.60**                        |
>
> After debiasing, MA-PPO still significantly outperforms PPO, confirming that its advantage is not driven by length bias.
>
> Regarding baseline training sufficiency, as shown in Figure 2, PPO and MA-PPO converge at the end of training. Extending training further led to overfitting and decreased reward scores for both methods. This suggests that the baselines were adequately trained.
>
> [1] Length-Controlled AlpacaEval: A Simple Way to Debias Automatic Evaluators https://arxiv.org/abs/2404.04475
>
> > (W2): The equation for R is incorrect in Section 2.2.
> > * KL is between two distributions, but the authors wrote that the KL is between two real numbers.
> > * The last term should be -KL( \pi_\theta ( \cdot | x) || \pi_\text{ref} (\cdot | x)); in particular, KL(p || q) does not equal to -KL(q || p). Currently if you expand the KL, the interpretation is that if \pi_sft (y|x) is large, then we should make \pi_\theta (y|x) really small, in order to maximize R.
> > Other issues in Section 2
> > * Can gamma be equal to 1? (Section 2.1)
> > * In Eq. (1) it’s unclear if A_t is computed using \theta or \theta_old – it’s never defined; similarly, how about Section 3 line 233 – what model would the authors use to compute the advantage?
> > * In Eq. (1) what’s the expectation over?
> > * At the end of page 2, the indices start from 0 (e.g., s_0, a_0); but on page 3 line 145, the indices start from 1
>
> **Re #W2**: Equation Issues in Section 2
> Thank you for pointing out the errors. We have revised the equation for $ R(x, y) $:
> $$ R(x, y) = r_\phi(x, y) - \beta D_{\mathrm{KL}}(\pi_\theta( \cdot \mid x) || \pi_{\mathrm{sft}}( \cdot \mid x)) $$
> This correction ensures consistency in the KL divergence term.
> Regarding other issues:
> 1. Can $\gamma$ be equal to 1? Yes, $\gamma = 1$ represents the no-discount scenario in RL.

---

> ### Author Response · Authors · 2024-11-22
> **Response to Reviewer QYzk (Part 2/2)**
>
> **Re #W2**:
>
> 2. Computation of $A_t$: The advantage function $A_t$​ is calculated using Generalized Advantage Estimation (GAE) [1], which combines the outputs of the critic model (values) and the reward model (rewards). In PPO [2], both the current policy $\pi_{\theta}$​ and the old policy $\pi_{\theta_\text{old}}$​​ are utilized for this computation. For rewards, a common approach assigns the holistic reward produced by the reward model to the last position of the response (EOS token), while token-level rewards such as KL divergence may also be used. However, when KL divergence is treated as a loss function rather than a reward signal, the computation of $A_t$ becomes less central to the optimization process. To simplify our exposition, we omitted the detailed computation of $A_t$​ in Section 2.1 when introducing policy optimization. For further details, please refer to the original works on GAE [1] and PPO [2].
>
> 3. Expectation in Eq. (1): This expectation $\mathbb{E}_t[\dots]$ indicates the empirical average over a finite batch of sample.
>
> 4. Indexing discrepancy: We have standardized the indices throughout the paper to start from 0.
>
> [1] High-Dimensional Continuous Control Using Generalized Advantage Estimation https://arxiv.org/abs/1506.02438
>
> [2] Proximal Policy Optimization Algorithms https://arxiv.org/abs/1707.06347
>
> **Response to Questions**:
>
> > (Q1): A natural way is based on BPE-like automatic construction of a dictionary with most frequent n-grams. Have the authors considered this strategy?
>
> **Re #Q1**: Thank you for your insightful feedback! While BPE-based construction could be insightful, it poses challenges in on-policy training. PPO requires sampling responses during training, whereas BPE construction relies on iterative merging operations with a large corpus of sampled response. Therefore, it is expensive to build and adapt a BPE dictionary dynamically during PPO training. However, it is indeed a promising direction for future work.
>
> > (Q2): Do the authors think that leave-one-out REINFORCE or other recent PPO variants would also benefit from the authors’ algorithm (macro-actions)?
>
> **Re #Q2**: Leave-one-out REINFORCE (RLOO) is typically applied to bandit problems, whereas MA-PPO bridges PPO and REINFORCE by balancing per-token and macro-action levels. Macro-actions could enhance other PPO variants by providing temporal abstraction and improving credit assignment, as demonstrated by our scalability and convergence speed improvements.
>
> To evaluate RLOO’s performance, we conducted experiments on the TL;DR dataset:
> | **Method**   | **RM Score** |
> |--------------|--------------|
> | SFT          | -0.64        |
> | PPO          | 0.83         |
> | MA-PPO (n=5) | **1.40**     |
> | RLOO (K=4)   | 0.81         |
>
> | ﻿                 | **Win** | **Tie** | **Loss** |
> |------------------|---------|---------|----------|
> | RLOO v.s. PPO    | 25      | 1       | 24       |
> | RLOO v.s. MA-PPO | 12      | 2       | 36       |
>
> RLOO achieved comparable performance to PPO but was outperformed by MA-PPO.
>
> **Training details for RLOO**: The learning rate for the policy model is set to 1.5e-5, and the number of online samples is K = 4. All other hyperparameters are kept consistent with PPO.
>
> We hope our responses clarify your concerns and address all points of feedback. Thank you again for your constructive review, and we look forward to further discussion.

---

> ### Author Response · Authors · 2024-11-26
> **Request for Re-review and Discussion**
>
> Dear Reviewer QYzk,
>
> We appreciate your constructive feedback! As the response period nears its end, we kindly ask if our responses and revisions have addressed your concerns. If so, we respectfully request you to reconsider your assessment. Further, we are happy to address any further concerns you may have!

---

> > ### Comment · Reviewer_QYzk · 2024-11-26
> >
> > Thank you for the response and the additional experiments.
> >
> > On BPE: I meant constructing it at the beginning (either using the training corpus or using some large corpus that looks similar to the training corpus), so each token might contain multiple words. We don't need to update the dictionary dynamically.
> >
> > I agree with reviewer ziZM that the term "macro-action" is slightly misleading, because the policy does not take macro-actions during online sampling -- the way it's taking actions is different from default/regular RL which readers are familiar with. I saw that the authors put a few lines in Limitations but clarifying this concept early in the paper would help. Speaking of limitations: LLM practitioners may often do RL on all sorts of prompts. Would macro-action hurt any task at all (e.g., math reasoning)?

---

> > > ### Author Response · Authors · 2024-11-28
> > >
> > > > On BPE: I meant constructing it at the beginning (either using the training corpus or using some large corpus that looks similar to the training corpus), so each token might contain multiple words. We don't need to update the dictionary dynamically.
> > >
> > > We appreciate the clarification regarding the construction of BPE tokens at the beginning using a static dictionary. Indeed, constructing the most frequent BPE pairs relies heavily on the specific corpus used, making the process inherently data-dependent. Additionally, since PPO training does not require annotated responses (in practice, only the prompts are needed) and instead samples training responses on-the-fly, relying on a predefined static vocabulary may limit flexibility during optimization. However, it is still a promising direction to explore for future work. Thank you again for your thoughtful comments!
> > >
> > >
> > > > I agree with reviewer ziZM that the term "macro-action" is slightly misleading, because the policy does not take macro-actions during online sampling -- the way it's taking actions is different from default/regular RL which readers are familiar with. I saw that the authors put a few lines in Limitations but clarifying this concept early in the paper would help.
> > >
> > > Thank you for your comments on the misunderstandings! We have provided clarifications on the macro action definitions and revised the manuscript accordingly. In our implementations, the macro actions is actually the "single-step" or primitive options (macro actions), as noted in the literature [1, 2]. We have made revisions in Line#218-222 and Line#159 to further clarify this point in our paper.
> > >
> > > > Speaking of limitations: LLM practitioners may often do RL on all sorts of prompts. Would macro-action hurt any task at all (e.g., math reasoning)?
> > >
> > > Our experiments comprehensive evaluated on four different tasks: text summarization, dialogue generation, question answering, and program synthesis, showing the effectiveness of our approach. Regarding additional evaluations, such as mathematical reasoning, we note that this task shares similarities with code generation, where our method has already shown strong performance. Therefore, we believe our approach is likely to generalize well to such tasks. However, due to the limited pages and time constraints, we were unable to extend our experiment beyond the four validated tasks. We thank the reviewer for raising this point and believe it is a promising direction to investigate for future work.
> > >
> > > We would like to thank the reviewer QYzk for your insightful feedback and engagement! We are happy to address them if there are any concerns requiring further discussion.
> > >
> > > **References**:
> > >
> > > [1] Sutton, Richard S., Doina Precup, and Satinder Singh. "Between MDPs and semi-MDPs: A framework for temporal abstraction in reinforcement learning." Artificial intelligence 112.1-2 (1999): 181-211.
> > >
> > > [2] McGovern, Amy, and Richard S. Sutton. "Macro-actions in reinforcement learning: An empirical analysis." Computer Science Department Faculty Publication Series (1998): 15.

---

> > > ### Author Response · Authors · 2024-12-02
> > > **Request for Further Discussion**
> > >
> > > Dear Reviewer QYzk,
> > >
> > > Thank you for your thoughtful comments! We hope that our detailed responses have addressed your concerns and clarified any outstanding questions.
> > >
> > > **As the extended discussion period approaches its conclusion, we respectfully request your engagement in further discussion and kindly ask you to reconsider your overall assessment of our submission.**
> > >
> > > Thank you again for your time and efforts in reviewing our work!

---

### Official Review · Reviewer_85YK · 2024-11-02

**Soundness:** 3
**Presentation:** 3
**Contribution:** 3
**Rating:** 8
**Confidence:** 4

**Summary:**

This paper introduces MA-RLHF, a novel framework that incorporates macro actions (sequences of tokens or higher-level language constructs) into the RLHF training process for LLMs. The key innovation is operating at a higher level of abstraction by grouping tokens into meaningful units, which helps address the credit assignment problem in long sequences. The authors demonstrate that this approach leads to faster convergence (1.7x to 2x) and better performance across various tasks including text summarization, dialogue generation, question answering, and program synthesis, without increasing computational complexity during training or inference.

**Strengths:**

- The paper identifies a critical limitation in token-level RLHF - the credit assignment problem over long sequences - and proposes a simple yet effctive solution by incorporating macro actions. The approach is well-motivated by both theoretical considerations (credit assignment, temporal abstraction) and practical issues (subword tokenization challenges).

- The paper is well-written and easy to follow.

- The experimental evaluation is thorough, covering multiple tasks, model sizes (2B to 27B parameters), and evaluation metrics. The authors also conducted detailed ablation studies on different strategy of termination and find n-gram performs the best, analyzed the impact of macro action size, and did best-of-N to further validate the effectiveness of the proposed approach.

- The proposed method is simple (maintaining the same computational complexity) yet effectively improve performance (up to 30% in text summarization and code generation). The faster convergence and better performance make this a valuable contribution to practical applications.

**Weaknesses:**

- While the paper explores different termination conditions for macro actions (n-gram based, perplexity based, parsing based), there could be more analysis of how different types of macro actions affect different types of tasks. For example, which macro action strategies work best for which types of generation tasks? For example, i would expect parsing-based termination might also work well on code generation tasks if a programming-language-based parser was used.

- Lack details of Human Evaluation: The evaluation involves human evaluation of model responses. It would be nice to show the detailed protocol of how human evaluation is performed, and what is the inter-rater agreement.

**Questions:**

* How sensitive is the method to the choice of macro action size? Is the macro action sizes sensitive to different types of tasks - i’m particularly interested in difference between different categories of tasks, for example, difference between natural language (TLDR, HH-RLHF done in the paper) and coding (APPS). In figure 6, it seems the difference between different n choice is not too large on HH-RLHF, wondering if that would change for coding tasks.

* Will be nice to see some qualitative examples of the macro-actions (e.g., how tokens are grouped together in a concrete setence).

* What is the difference & commonalities between this paper and ArCHer [1]?
[1] ArCHer: Training Language Model Agents via Hierarchical Multi-Turn RL

---

> ### Author Response · Authors · 2024-11-22
> **Response to Reviewer 85YK (Part 1/3)**
>
> Dear Reviewer 85YK,
>
> Thank you for your thoughtful review and positive feedback. Below, we address your comments and questions in detail:
>
> **Response to Weaknesses**:
>
> > (W1): While the paper explores different termination conditions for macro actions (n-gram based, perplexity based, parsing based), there could be more analysis of how different types of macro actions affect different types of tasks. For example, which macro action strategies work best for which types of generation tasks? For example, i would expect parsing-based termination might also work well on code generation tasks if a programming-language-based parser was used.
>
> **Re #W1**: We appreciate your suggestion to explore how different macro action termination strategies perform across tasks. As shown in Figure 5, we evaluated termination strategies (fixed, parsing, perplexity) on the TL;DR dataset and found fixed termination sufficient. To further analyze task-specific behaviors, we conducted additional experiments on the HH-RLHF and APPS datasets.
>
> **For HH-RLHF**:
>
> We compared parsing-based termination with fixed n-gram-based termination. Results evaluated by both RM and GPT-4 are as follows:
>
> | **Method**       | **RM Score (HH-RLHF)** |
> |------------------|------------------------|
> | SFT              | 0.13                   |
> | PPO              | 1.31                   |
> | MA-PPO (n=5)     | 1.55                   |
> | MA-PPO (Parsing) | **1.64**               |
>
> | **HH-RLHF**                        | **Win** | **Tie** | **Loss** |
> |------------------------------------|---------|---------|----------|
> | MA-PPO (Parsing) v.s. MA-PPO (n=5) | 30      | 6       | 14       |
>
> Parsing-based termination performed particularly well on dialogue tasks, likely due to its ability to model structured and coherent responses.
>
> **For APPS (code generation)**:
>
> We implemented a programming-language-based AST parser (using the RedBaron toolkit https://github.com/PyCQA/redbaron) and compared it with other termination strategies. Results are summarized below:
>
> | **Macro Action Termination** | **Interview** | **Introductory** | **Competition** | **All**  |
> |------------------------------|---------------|------------------|-----------------|----------|
> | Vanilla PPO                  | 2.82          | 15.26            | 0.92            | 4.92     |
> | Fixed 10-gram                | **3.25**      | 16.56            | 0.94            | 5.45     |
> | Parsing                      | 3.17          | **17.05**        | **1.24**        | **5.56** |
> | Perplexity                   | 3.04          | 16.36            | 0.80            | 5.26     |
>
> Parsing-based termination consistently outperformed others, as it captured the structural patterns inherent in code. Perplexity-based termination, however, underperformed due to fine-grained and style-insensitive macro actions.
>
> > (W2): Lack details of Human Evaluation: The evaluation involves human evaluation of model responses. It would be nice to show the detailed protocol of how human evaluation is performed, and what is the inter-rater agreement.
>
> **Re #W2**: The human evaluation protocol is detailed in Appendix E.2. When conducting human evaluation, the annotators are given an instruction and two responses, they should choose which response is better than the other based on certain task-specific aspects. We illustrate these task-specific evaluation aspects as follows:
>
> **Summarization Task**:
>
> * **Hallucination**: this considers whether the generated summary includes any additional information not present in the original post or article.
>
> * **Verbosity**: this assesses if the summary includes unnecessary context that could be removed without negatively impacting its quality.
>
> * **Overall Quality**: this measures the general coherence, informativeness, and readability of the generated summary.
> Evaluation Protocol of Summarization:
> The annotators should first compare the overall quality of two responses. If overall qualities are equally good for responses, then they should choose the winner based on hallucination and verbosity.
>
> **Dialogue Task**:
> * **Instruction Following**: whether the generated response follows the requirements in the instruction.
> * **Usefulness**: whether the advices in the response are applicable, and does the response ideally guide the user on what to do next.
>
> The annotators should choose the winner by comprehensively considering these two aspects.
>
> Each response pair was evaluated by three annotators, with inter-rater agreements as follows:
> |                   | **TL;DR** | **HH-RLHF** | **Avg.** |
> |-----------------------|-----------|-------------|----------|
> | Inter-rate  agreement | 64%       | 72%         | 68%      |

---

> ### Author Response · Authors · 2024-11-22
> **Response to Reviewer 85YK (Part 2/3)**
>
> **Response to Questions**:
>
> > (Q1): How sensitive is the method to the choice of macro action size? Is the macro action sizes sensitive to different types of tasks - i’m particularly interested in difference between different categories of tasks, for example, difference between natural language (TLDR, HH-RLHF done in the paper) and coding (APPS). In figure 6, it seems the difference between different n choice is not too large on HH-RLHF, wondering if that would change for coding tasks.
>
> **Re #Q1**: We believe the sensitivity to macro-action size is task-specific. As shown in our paper, we varied the macro-action size on TL;DR and HH-RLHF datasets. For TL;DR, a macro-action size of 5 performed best, while for HH-RLHF, a size of 10 yielded the best results. This difference may be attributed to the average length of responses, as HH-RLHF responses are generally longer than those in TL;DR. However, these language generation tasks are less sensitive to the macro action size.
>
> For coding task, we conduct experiments with 3 different sizes: 3, 10, and 20. Results are shown below:
>
> | **Macro Action Size** | **Interview** | **Introductory** | **Competition** | **All**  |
> |-----------------------|---------------|------------------|-----------------|----------|
> | Vanilla PPO           | 2.82          | 15.26            | 0.92            | 4.92     |
> | Fixed 3-gram          | 3.08          | 16.16            | 0.94            | 5.27     |
> | Fixed 10-gram         | **3.25**      | 16.56            | 0.94            | **5.45** |
> | Fixed 20-gram         | 3.14          | **16.83**        | **0.97**        | 5.44     |
>
> We notice that when macro action size is less than 10, the MA-PPO is less effective, but when it is large or equal than 10, the performance is relative stable. From our observation of samples (which are listed in the response to Q2), we find that a smaller macro action size fails at capturing relavent action together, while size = 10 is large enough for capturing the structure of code.
>
> > (Q2): Will be nice to see some qualitative examples of the macro-actions (e.g., how tokens are grouped together in a concrete setence).
>
> **Re #Q2**: Here, we provide a qualitative example under different terminations, and we use the @@ symbol to separate the macro actions:
>
> | Termination                    | Macro Actions                                                                                                                                                                                    |
> |--------------------------------|--------------------------------------------------------------------------------------------------------------------------------------------------------------------------------------------------|
> | (Original Response)            | 20M, moved to Europe, met some people but they quickly disappeared, want to celebrate, but I'm still alone, would it seem weird or desperate if I go out alone?                                  |
> | Fixed 5-gram based termination | 20M, moved@@ to Europe, met some@@ people but they quickly disappeared@@, want to celebrate,@@ but I'm still@@ alone, would it seems@@ weird or desperate if I@@ go out alone?@@                 |
> | Perplexity-based termination   | 20M, moved to@@ Europe@@,@@ met some people@@ but they@@ quickly disappeared@@, want@@ to celebrate,@@ but I'm still@@ alone@@,@@ would it seems@@ weird or desperate if@@ I go out@@ alone@@?@@ |
> | Parsing-based termination      | 20M,@@ moved to Europe,@@ met some people but they quickly disappeared,@@ want to celebrate, but@@ I'm still alone, would it seems@@ weird or desperate if@@ I go out alone?@@                   |
>
> This example shows that given different terminations, they are sufficient enough to include structures into a macro action, since the sentence does not involve with complicated response in it.

---

> ### Author Response · Authors · 2024-11-22
> **Response to Reviewer 85YK (Part 3/3)**
>
> **Re #Q2**: Also, we present an example from code task:
>
> **(Original Response)**:
> ```python
> def queue_time(customers, n):
>     tills = [0] * n
>     for i in range(len(customers)):
>         tills[i % n] += customers[i]
>     return max(tills)
> ```
> **Fixed 3-gram based termination**:
> ```python
> def queue_@@time(customers@@, n):@@
>     till@@s = [@@0] *@@ n
>     @@for i in@@ range(len@@(customers)):@@
>         till@@s[i@@ % n]@@ += customers[@@i]
> @@    return max@@(tills@@)@@
> ```
> **Fixed 10-gram based termination**:
> ```python
> def queue_time(customers, n):
> @@    tills = [0] * n
> @@    for i in range(len(customers)):@@
>         tills[i % n] +=@@ customers[i]
>     return max(till@@s)@@
> ```
> **Perplexity-based termination**:
> ```python
> def queue@@_@@time@@(customers@@, n@@):
> @@    tills@@ = [@@0]@@ *@@ n
> @@    @@for i@@ in range@@(len@@(customers@@)):
> @@        tills@@[@@i % n@@] +=@@ customers[@@i@@]@@
>     return max@@(till@@s@@)@@
> ```
> **Parsing-based termination**:
> ```python
> def queue_time(customers, n):@@
>     @@tills = [0] * n@@
>     @@for i in range(len(customers)):@@
>         @@tills[i % n] += customers[i]@@
>     @@return max(tills)@@
> ```
> From this example, we can find that both 3-gram and perplexity based terminations fail to capture the structure of code, while 10-gram and parsing based terminations otherwise. The less effectiveness of 3-gram and perplexity based terminations could be attributed to this.
>
> > (Q3): What is the difference & commonalities between this paper and ArCHer [1]? [1] ArCHer: Training Language Model Agents via Hierarchical Multi-Turn RL
>
> **Re #Q3**: ArCHer introduces a hierarchical Markov decision process (MDP) with high-level (utterance-level) and low-level (token-level) policies for multi-turn RL. In contrast, MA-RLHF uses macro actions to transform the token-level MDP into a single-level macro-action MDP.
>
> The commonality lies in leveraging higher-level abstractions for decision-making, while the key difference is the scope of application: ArCHer targets multi-turn dialogues, while MA-RLHF applies macro actions broadly to diverse tasks, including summarization and code generation.
>
> We hope these responses clarify your questions and concerns. Thank you for your constructive feedback and for recognizing the value of our work!

---

> > ### Comment · Reviewer_85YK · 2024-11-26
> >
> > I appreciated the detailed response and additional experiments that reveal interesting findings - hopefully they will be included in the full paper. I will keep my score and vote for acceptance :)

---

### Official Review · Reviewer_YfQu · 2024-11-04

**Soundness:** 3
**Presentation:** 3
**Contribution:** 3
**Rating:** 8
**Confidence:** 3

**Summary:**

They conduct empirical analysis of the use of macro actions in RLHF instead of the standard per-token actions. They study a variety of approaches to breaking responses into macro actions, finding that N-gram based macro actions tend to perform best. In general, they find macro actions have the potential to outperform standard PPO with per token actions. They attribute this to better training stability and reduced variance in the learning problem.

**Strengths:**

* They conduct thorough analysis on many different tasks, using several different sized base models.
* They conduct many ablations of their method.
* The paper is clearly written and easy to follow.
* This empirical analysis of macro actions present a useful contribution to our understanding of RLHF training.

**Weaknesses:**

* Some of the differences in performance could be due to tuning one method more than another. It would be great if the paper could at least document the steps they went through to tune hyper-parameters for their method and baselines.
* My understanding is that the primary functional difference between per-token PPO and macro action PPO is the granularity of the value function, importance sampling, and discount factor. It is a fairly small modification, but this is not necessarily clear when reading the paper and could be better emphasized.
* The GRPO method proposed in https://arxiv.org/pdf/2402.03300, can be viewed as a special case of their macro action PPO. They don't discuss this in the related work.

**Questions:**

See weaknesses for questions.

---

> ### Author Response · Authors · 2024-11-22
> **Response to Reviewer YfQu**
>
> Dear Reviewer YfQu,
>
> Thank you for taking the time to review this paper. Here are our responses and clarifications regarding your questions:
>
> > (W1) Some of the differences in performance could be due to tuning one method more than another. It would be great if the paper could at least document the steps they went through to tune hyper-parameters for their method and baselines.
>
> **Re #W1**: In our experiments, we did not perform separate hyperparameter tuning specifically for MA-PPO. Instead, we first tuned the hyperparameters for vanilla PPO on each dataset and then applied the same settings to train MA-PPO.
>
> Adjustments were only made when convergence issues arose, and such changes were applied equally to both methods. For instance, when training 7B models, we began with policy and critic learning rates of 5e-6 and gradually reduced them to 1e-6. Similarly, for 27B models, the policy learning rate was reduced from 1e-6 to 7e-7, the KL coefficient was increased from 0.05 to 0.1, and warmup steps were reduced from 200 to 0.
>
> To ensure clarity and reproducibility, we have reported the key hyperparameter settings for PPO and MA-PPO across all experiments in Appendix A.2. Here are some importance hyper-parameters that we used when training PPO and MA-PPO:
>
> | **Hyper-Parameter**  | **Gemma 2B** | **Gemma 7B**                    | **Gemma 27B** | **CodeGemma 2B** | **CodeGemma 7B** |
> |----------------------|--------------|---------------------------------|---------------|------------------|------------------|
> | Policy learning rate | 1.5e-5       | 1e-6                            | 7e-7          | 5e-7             | 5e-7             |
> | Critic learning rate | 1.5e-5       | 1e-6                            | 1e-6          | 5e-5             | 5e-5             |
> | Epochs               | 1            | 1                               | 1             | 1                | 1                |
> | $\lambda$ in GAE     | 0.95         | 0.95                            | 0.95          | 0.95             | 0.95             |
> | $\gamma$ in GAE      | 1            | 1                               | 1             | 1                | 1                |
> | KL coefficient       | 0.05         | 0.1 for WebGPT | 0.1           | 0.05             | 0.05             |
> | | | 0.05 for others | | | | |
> | Warmup steps         | 200          | 200                             | 0             | 20               | 20               |
>
> > (W2): My understanding is that the primary functional difference between per-token PPO and macro action PPO is the granularity of the value function, importance sampling, and discount factor. It is a fairly small modification, but this is not necessarily clear when reading the paper and could be better emphasized.
>
> **Re #W2**: Thanks for your insightful comment. The core modification of integrating macro-action into PPO primarility affects the calulations of the value function, importance sampling, and discount factor. In the revised manuscript, we have highlighted this **at the end of Section 3.2.1**, after illustrating the termination condition, where we point out the corresponding modifications on value function and importance sampling, and use the simple averaging strategy to rearrange them. Besides, we also disscus the different strategies that could be used in here in Appendix C.1, and refer to it in the paper.
>
> > (W3): The GRPO method proposed in https://arxiv.org/pdf/2402.03300, can be viewed as a special case of their macro action PPO. They don't discuss this in the related work.
>
> **Re #W3**: We appreciate your suggestion to discuss with GRPO. GRPO evaluates rewards at each reasoning step, making it a special case of MA-PPO where each reasoning step is treated as a macro action. However, GRPO is tailored for math-related problems, while MA-PPO is designed for broad applicability across diverse tasks such as summarization, dialogue, and program synthesis.
>
> In the updated manuscript, we have included a discussion in the related work section to clarify this relationship and highlight the broader scope of our approach.
>
> Thank you again for your valuable feedback, and we are grateful for your positive assessment of our work!

---

> > ### Comment · Reviewer_YfQu · 2024-11-26
> >
> > Thank you for responding to my concerns! I commend the authors for the additional clarifications they added. I will keep my score as I think this is a solid paper.

---

### Official Review · Reviewer_oNFY · 2024-11-05

**Soundness:** 2
**Presentation:** 3
**Contribution:** 2
**Rating:** 6
**Confidence:** 3

**Summary:**

The current manuscript proposes to tackle the credit assignment problem when aligning large language models (LLMs) to human preferences via reinforcement learning from human feedback (RLHF). The core idea is to consider a group of tokens as a macro action, instead of individual actions, and use rewards accumulated within a macro action with proximal policy optimization (PPO) to align the language model. This approach is labeled MA-RLHF. Experiments on three tasks (text summarization, dialog generation, question answering, and program synthesis) show improvements of MA-RLHF over PPO for a chosen LLM (Gemma) both in terms of performance and training speed.

**Strengths:**

(S1) The direction of the current work to use macro actions (or a group of low-level actions together) is novel and interesting, for the setting of aligning a language model with human preferences (RLHF scope). Please see W1 for more discussion.

(S2) The manuscript is well-written, with sufficient details on all the technical details making it easy to understand and comprehend. I thoroughly enjoyed reading the paper.

(S3) The ablation and generalization studies presented in the experiment section (Sec.4.3 and 4.4) were interesting to better understand the behavior of the proposed approach in the context of RLHF.

**Weaknesses:**

(W1) The idea of gathering multiple actions together as a “macro-action” is not novel in general. Referred to as “action chunking” [A], similar ideas have been explored in other domains. As noted in (S1), this is still useful in the context of RLHF, but with a diminished novelty factor.

(W2) The current work mostly experiments with a single LLM (Gemma and its variants). Given (W1) and lack of generalizability across different models, the usefulness of the current approach over PPO is not clearly understood to benefit the community. Demonstrating similar benefits in an LLM-agnostic manner will tremendously be useful.

(W3) Comparisons to the more recent approaches like DPO [B] are missing in the experimental tables. Though there is benefit over PPO for the chosen model, the experiments fall short to gauge the improvements compared to one of the current arts. Request the authors to include this comparison to better understand the proposed approach and its efficacy.

References
* [A] Action chunking as conditional policy compression. https://osf.io/preprints/psyarxiv/z8yrv
* [B] Direct Preference Optimization: Your Language Model is Secretly a Reward Model.
https://osf.io/preprints/psyarxiv/z8yrv

**Questions:**

Typos:
L68: leads -> lead

---

> ### Author Response · Authors · 2024-11-22
> **Response to Reviewer oNFY**
>
> Dear Reviewer oNFY,
>
> We sincerely appreciate your thoughtful review and the constructive feedback. Below, we address your concerns in detail.
>
> **Response to Weaknesses:**
>
> > (W1) The idea of gathering multiple actions together as a “macro-action” is not novel in general. Referred to as “action chunking” [A], similar ideas have been explored in other domains. As noted in (S1), this is still useful in the context of RLHF, but with a diminished novelty factor.
>
> **Re #W1**: We acknowledge that macro actions have been explored in other domains, such as "action chunking" [A]. However, our work fundamentally differs in the following ways:
>
> 1. **Domain Difference**: [A] applies action chunking in instrumental learning tasks for cognitive behavior studies, whereas our work focuses on macro actions in RLHF for LLM alignment and text generation.
>
> 2. **Methodological Difference**: [A] uses a learned system for macro actions, while we employ predefined rule-based heuristics to accelerate PPO alignment in LLM training.
>
> As the first study to apply macro actions in RLHF for LLMs, we believe our work retains significant novelty and respectfully encourage the reviewer to reconsider this aspect.
>
> > (W2) The current work mostly experiments with a single LLM (Gemma and its variants). Given (W1) and lack of generalizability across different models, the usefulness of the current approach over PPO is not clearly understood to benefit the community. Demonstrating similar benefits in an LLM-agnostic manner will tremendously be useful.
>
> **Re #W2**:  Thank you for your comments. To address the reviewer's concern regarding the LLM-agnostic experiments, we further conduct experiments on Llama-3.2-3B, to validate the generalizability of our method. We conduct experiments on TL;DR datasets, following the same data split as Gemma-2B. We set the learning rates of actor and critic to 5e-6 and 1e-5, and the KL coefficient is set to 0.1. We report our RM score and GPT-4 assessment results below:
>
> | **Method**   | **RM Score (TL;DR)** |
> |--------------|----------------------|
> | SFT          | 2.38                 |
> | PPO          | 3.33                 |
> | MA-PPO (n=5) | **3.96**             |
>
> | **TL;DR dataset** | **Win** | **Tie** | **Loss** |
> |-------------------|---------|---------|----------|
> | MA-PPO v.s. PPO   | 31      | 17      | 2        |
>
> These results validate that MA-PPO consistently outperforms baselines, demonstrating its LLM-agnostic benefits. We hope this addresses your concerns regarding generalizability. The results have been included in the manuscript Appendix B.4.
>
> > (W3) Comparisons to the more recent approaches like DPO [B] are missing in the experimental tables. Though there is benefit over PPO for the chosen model, the experiments fall short to gauge the improvements compared to one of the current arts. Request the authors to include this comparison to better understand the proposed approach and its efficacy.
>
> **Re #W3**:  Thank you for your suggestions on reporting DPO results. Since our methods mainly focus on RL-based alignment and it is increasing known that DPO is not as good as PPO in recent literature, we only reported the RL-based results as our baselines. However, to address the reviewer's concerns, we provide the DPO comparisons below, evaluated with both RM and GPT-4:
>
> **Reward Model Scores**:
> | **Method**   | **RM Score (TL;DR)** | **RM Score (HH-RLHF)** |
> |--------------|----------------------|------------------------|
> | SFT          | -0.64                | 0.13                   |
> | PPO          | 0.83                 | 1.31                   |
> | MA-PPO (n=5) | **1.40**             | **1.55**               |
> | DPO          | 0.03                 | 0.64                   |
>
> **Win/Tie/Loss Comparisons (GPT-4)**:
>
> _TL;DR Dataset_:
>
> | **TL;DR**       | **Win** | **Tie** | **Loss** |
> |-----------------|---------|---------|----------|
> | DPO v.s. PPO    | 17      | 5       | 28       |
> | DPO v.s. MA-PPO | 4       | 6       | 40       |
>
> _HH-RLHF Dataset_:
>
> | **HH-RLHF**     | **Win** | **Tie** | **Loss** |
> |-----------------|---------|---------|----------|
> | DPO v.s. PPO    | 26      | 2       | 22       |
> | DPO v.s. MA-PPO | 21      | 4       | 25       |
>
> DPO underperforms compared to both PPO and MA-PPO on TL;DR, while  DPO exhibits a higher win rate compared to PPO but still performs worse than MA-PPO on HH-RLHF. We believe that these additional experimental results adequately address your concerns. Also, the results are added in the latest manuscript Appendix B.3.
>
> **Training Details of DPO**: The learning rate is set to 2e-7, with β = 0.1 for TL;DR and β = 0.01 for HH-RLHF. The policy and reference models are initialized using the same SFT model as in PPO.
>
> ---
> We hope these additional experiments and clarifications address your concerns and kindly encourage you to reconsider your score in light of the updates! We are grateful for the opportunity to engage further.

---

> ### Author Response · Authors · 2024-11-25
> **Request for Re-review and Discussion**
>
> Thank you for your constructive feedback! As the official response period approaches its conclusion (November 26 at 11:59pm AoE), we kindly request you to reconsider our assessment score in light of our responses and revisions.
>
> If you have any remaining concerns or questions, we would be happy to engage in further discussion before the response period ends. We greatly appreciate your time and consideration!

---

> ### Comment · Reviewer_oNFY · 2024-11-26
> **Post-Rebuttal Response**
>
> Thanks to the authors for their response to the review.
>
> After going through it along with other reviews, I believe there is merit in the current work and am raising my rating.

---

### Author Response · Authors · 2024-11-22
**Summary of Revisions**

Dear Reviewers and AC:

We sincerely thank you for your valuable time and constructive feedback. We deeply appreciate the recognition of our method's novelty, technical contributions, and clarity, as highlighted by Reviewers oNFY, YfQu, 85YK, and QYzk. Your acknowledgment of our comprehensive ablation and generalization studies, along with the robustness of our experiments across various tasks and models, is highly encouraging. Additionally, we are grateful for the detailed suggestions regarding experiments (Reviewers oNFY, 85YK) and writing improvements (Reviewers YfQu, QYzk, ziZM), which have helped us refine and extend our contributions.
To address the raised concerns, we have implemented several key revisions, summarized below:

1. For Reviewer oNFY, ziZM, we have fixed the typos mentioned in your comments.

2. For Reviewer QYzk, ziZM, we modify the equation in Section 2.2 related to reward function to the correct format.

3. For Reviewer oNFY, we include the experimental results of DPO baseline in Appendix B.3, experiments on Llama-3.2-3B in Appendix B.4.

4. For Reviewer QYzK, we revise the equation (1) with the expectation over, and change the indices and now start from 0 in Line#148, corresponding to the former equation.

5. For Reviewer YfQu, we have highlighted the modification that we made when implementing MA-PPO at the end of Section 3.2.1, from Line#213 to Line#215. Besides, we discuss the GRPO in related work in Section 5, Line#520-521.

6. For Reviewer 85YK, we supplement the details of human evaluation in Appendix E.2.

7. For Reviewer ziZM, we include the results with applying different termination condition on various tasks in Appendix C.2.

8. For Reviewer ziZM, we update the Figure 2, 4, and Figure 10 to address the confusing that has been made.

9. For Reviewer ziZM, we use $r_t$ instead of $r(s_t, a_t)$ or $r(s_t)$ for consistency. The acronyms of SFT and RM are now explicitly expressed. We also change the order of introducing RM, introcude preference dataset first and then introduce the formula of RM. We delete the symbol $r$ stands for the likelihood ratio for better understanding.

10. For Reviewer ziZM, we update the cite related to credit assignment problem in Line#13,45, and REINFORCE in Line#249. We revise some statements related to our motivation in Line#54-55, 66-67.

We believe these revisions address the majority of the concerns raised and further strengthen the manuscript. We are open to continued discussion and sincerely hope that our responses and revisions adequately address your concerns. We would greatly appreciate it if you could consider revising the ratings in light of the improvements. Thank you again for your valuable feedback and engagement!

---

### Author Response · Authors · 2024-11-25
**Request for Re-review and Discussion**

Dear Reviewers,

Thank you for your constructive feedback! As the **official response period** nears its conclusion (**November 26 at 11:59pm AoE**), we hope that our detailed responses, additional experimental results, and manuscript revisions have addressed your concerns and clarified the key aspects of our work.

We kindly encourage reviewers **oNFY**, **QYzk**, and **ziZM** to reconsider their initial assessments in light of the updates we provided, including the additional experiments and explanations tailored to your specific comments. We sincerely hope that our efforts demonstrate the robustness and significance of our contributions.

To summarize, our key contributions are as follows:

1. **Introducing Macro Actions into RLHF for LLMs**:
We are the first to integrate macro actions into the RLHF framework for large language models (LLMs), introducing three distinct macro action termination strategies—$n$-gram-based, perplexity-based, and parsing-based. By grouping adjacent language tokens into a coarser action, our method reduces the temporal distance between actions and final rewards, thereby improving the credit assignment process while maintaining computational efficiency.

2. **Comprehensive Experiments and Evaluations**:
Across diverse tasks—including *text summarization, dialogue generation, question answering*, and *program synthesis*—we demonstrate that MA-RLHF achieves superior alignment quality, as evaluated by reward models, GPT-4, and human annotators. Furthermore, MA-RLHF demonstrates significantly faster learning efficiency than vanilla RLHF, as evidenced by reward scores during training.

3. **Generalizability and Robustness Analysis**:
MA-RLHF generalizes effectively across model sizes (2B, 7B, and 27B parameters) and model families. Additionally, we demonstrate its robustness under varying sampling temperatures and best-of-N (rejection) sampling setups, consistently outperforming vanilla RLHF.

We believe our work contributes to improving the learning efficiency and optimization of RLHF by leveraging heuristic temporal abstractions. Chunking sequences into macro actions reduces decision resolution and results in performance gains that are well-supported by prior RL literature. Additionally, we are committed to supporting the open-source community and fostering further research by planning to release our code, datasets, and trained model checkpoints.

If there are any remaining points of clarification or concerns, we would be happy to address them promptly before the discussion phase ends. We hope these updates address your concerns and greatly appreciate your reconsideration!

Thank you very much!

Authors

---

### Meta-Review · Area_Chair_YZHQ · 2024-12-21

**Metareview:**

The paper introduces a framework that incorporates macro actions—sequences of tokens or higher-level language constructs—into the RLHF process for large language models (LLMs). This approach aims to address the credit assignment problem in token-level RLHF, where delayed rewards over long sequences hinder learning efficiency.  Empirical results demonstrate significant performance improvements across various tasks, including text summarization, dialogue generation, question answering, and program synthesis. There are some concerns (which the authors addressed well during the rebuttal) including application to more LLMs, comparison with DPO/PPO, training parameters in experiments, macro action definitions, etc. Please address these comments in the final paper.

**Additional Comments On Reviewer Discussion:**

n/a

---

### Decision · Program_Chairs · 2025-01-22

Accept (Poster)